# Fronto-motor circuits linked to effort-based decision-making and apathy in healthy subjects
Gerard Derosiere [1,2] ✉, Pierre Vassiliadis[3], Laurence Dricot[1], Quentin Dessain[1], Nicolas Delinte [1], Alexandre Zénon [4] & Julie Duque[1]

Apathy is marked by disruptions in effort-based decision-making for rewards and alterations in fronto–basal ganglia (BG) circuits. While changes in primary motor cortex (M1) activity during effort and reward valuation have been reported, prior work on apathy and effort-based decision-making has overlooked the connections between fronto–BG structures and M1. Here, we addressed this gap by investigating structural and effective connectivity within fronto-M1, fronto-BG-M1, and intra-M1 circuits in 45 healthy participants using tractography and paired-pulse transcranial magnetic stimulation. These measures were related to apathy scores and computational parameters of effort and reward valuation modelled from decision-making behavior. Apathy scores were associated with both structural and effective connectivity in orbitofrontal cortex-originating circuits. Additionally, effort and reward valuation were preferentially linked to supplementary motor area-related circuits and intra-M1 GABAergic circuits, respectively. These findings reveal dissociable circuit-level contributions to different dimensions of motivated behavior and identify potential neuromodulation targets for treating apathy.

Apathy, a syndrome marked by a diminished willingness to exert efforts for rewards[1-11], is highly prevalent in many brain disorders. It is found in 47% of schizophrenia cases, 61% of traumatic brain injury cases, 40% of Parkinson's Disease cases and up to 72% of frontotemporal dementia cases[12,13]. This prevalence highlights the need for research into the neural bases of apathy, to understand its biology, develop biomarkers, and identify circuits for targeted treatments[1,10]. Notably, studies of apathy in clinical populations can be confounded by neural changes associated with compensatory mechanisms or the pathology itself and often rely on small patient cohorts classified as either apathetic or non-apathetic, limiting their power[14]. A promising approach to identify the brain circuits underlying apathy is to study continuous apathy levels in larger groups of healthy individuals[15-17]. Interestingly, apathy levels vary significantly even among healthy individuals, with some showing high apathy scores[16-20]. Despite this, the neurobiology of apathy levels in the general population remains largely unexplored.

It has become clear that apathy is a multidimensional syndrome[21,22], which is not attributable to a single brain region but can rather be seen as a network disorder[13,17,23-27]. Whether clinical or subclinical[16,17], apathy is tied to alterations, consisting predominantly of decreases[13] though occasionally of increases, in the volume and activity of distributed fronto-basal ganglia (BG)

structures known for their role in deciding to exert efforts for rewards[13,23,24]. These structures include the supplementary motor area (SMA)[17] and the putamen[28], involved in effort valuation[29-31], the orbitofrontal cortex (OFC)[32], engaged in reward valuation[33,34], and the nucleus accumbens[35,36], thought to weigh up efforts and rewards[23,37]. Alterations are likewise observed in downstream BG nuclei, such as the globus pallidus[21], where effort and reward computations have also been reported[38,39]. Beyond these grey matter alterations, recent studies show that apathy is associated with altered white matter connectivity between fronto-BG structures[17,25-27], though the direction of these changes – whether increases or decreases – varies across studies, likely reflecting heterogeneity in apathy severity, study populations, or the specific circuits examined (see ref. 40, for review). This pattern suggests that, relative to individuals without apathy, both hypo- and hyper-connectivity in different components of the fronto-BG network can be associated with apathy. Such findings begin to outline what may constitute the apathy "dysfunctome"[41] – i.e., the set of connections altered in apathy that may be tuned by targeted treatments.

Strikingly though, while theories of distributed decision-making and embodied cognition increasingly recognize the crucial role of the primary motor cortex (M1) and its interactions with other brain regions[42-44],

[1]Institute of Neuroscience, Université Catholique de Louvain, Brussels, Belgium. [2]Université Claude Bernard Lyon 1, CNRS, INSERM, Centre de Recherche en Neurosciences de Lyon (CRNL), U1028 UMR5292, Impact Team, Bron, France. [3]Defitech Chair of Clinical Neuroengineering, Neuro-X Institute, École Polytechnique Fédérale de Lausanne (EPFL), Geneva and Sion, Switzerland. [4]INCIA, CNRS, Bordeaux, France. ✉e-mail: gerard.derosiere@inserm.fr

previous work on apathy has largely overlooked the brain's output structures, potentially missing key circuits in the apathy network. Yet, activity changes related to effort and reward valuation during decision-making have been observed repeatedly in M1 in rodents[45,46], non-human primates[47] and humans[48–52], suggesting a potential functional link between the fronto-BG structures mentioned above and M1 that may be central to effort-based decisions[53] and, presumably, to apathy.

In the past two decades, progress in neuroimaging and brain stimulation have enabled new ways to explore this idea in humans. Advanced analyses of diffusion-weighted imaging (DWI) such as streamline tractography allow quantifying how structural connectivity in specific circuits varies among individuals[54,55]. Additionally, paired-pulse transcranial magnetic stimulation (ppTMS) allows pre-activating certain circuits and measuring the impact of this pre-activation on neural activity in M1's corticospinal neurons[44,56,57], providing a measure of how effective connectivity in circuits projecting to these neurons varies among individuals. As such, recent studies have begun combining DWI and TMS to investigate cognitive processes[58]. Building on this emerging multimodal framework, integrating these complementary approaches provides a unique opportunity to comprehensively investigate the circuits connecting SMA, OFC, and the BG to M1.

In this study, we aimed to characterize how structural and effective connectivity within fronto-striato-motor circuits relate to individual differences in apathy in the general population. Specifically, we asked whether distinct circuits are differentially associated with apathy levels, as measured through questionnaires[17,18], and the valuation of efforts and rewards, as measured through computational phenotyping of behavior in an effort-based decision-making task[6,10,11,59]. While previous studies have linked apathy to altered connectivity within fronto-BG networks, findings have been mixed in terms of directionality – reporting both increases and decreases in connectivity – and largely focused on non-motor circuits. In light of these mixed findings and the paucity of data on motor circuits[17], the present study hypothesized altered connectivity within these circuits, without making directional assumptions regarding connectivity–behavior relationships. To address these questions, we combined DWI and dual-coil ppTMS to probe both structural and effective connectivity in key fronto-M1 and fronto-BG-M1 circuits originating from the SMA and OFC. In addition, single-site ppTMS was used to assess the physiological properties of intra-M1 circuits influencing corticospinal output. This multimodal framework thus allowed us to capture connectivity patterns across both anatomical and effective levels, from frontal input to motor cortex, and extending to the final corticospinal output stage. Among the most robust findings, we observed that both OFC-BG structural connectivity and OFC-M1 long-latency effective connectivity were positively associated with apathy scores, thus replicating this relationship across our two connectivity measures. Moreover, SMA-M1 structural connectivity was related to both apathy and effort valuation, consistent with SMA's known role in effort processing[29,30]. Altogether, these findings suggest that partially distinct fronto-motor circuits – originating from both OFC and SMA – contribute to apathy and effort valuation, highlighting complementary circuits as potential neuromodulatory targets for specific mechanisms of disrupted motivated behavior.

## Results

The study involved 45 healthy right-handed participants (25.1 ± 0.8 years old, 31 females, 14 males). We conducted a comprehensive neuropsychological assessment that included apathy scores (presented in Fig. 1A), as well as control variables related to apathy, such as depression and anhedonia scores (presented in Supplementary Fig. 1). Apathy scores ranged between 0.54 and 1.92 (Fig. 1A), consistent with previous studies using the same scale in healthy individuals[17,60,61], namely the extended version of the Lille Apathy Rating Scale (LARS-e)[17]. Higher scores indicated stronger apathy.

A possible limitation of apathy scales is that they rely on subjective psychological constructs[10]. To address this, we employed an effort-based decision-making task, where participants decided whether to exert varying

levels of effort with their right arm (biceps contraction) to obtain monetary rewards (see Fig. 1B and *Methods, Effort-based decision-making task*). Briefly, on each trial, participants were presented with one of 16 possible combinations of effort and reward levels, defined by 4 effort levels (20%, 40%, 60%, or 80% of their maximal voluntary contraction, MVC) and 4 reward amounts (1, 5, 10, or 20 euro cents). Using a keyboard, they had 5 seconds to decide whether to accept or reject the offer. If they accepted, they performed an isometric contraction of the right biceps, monitored via electromyography (EMG), and received the reward if the contraction met both intensity and duration criteria (see *Methods* for full details). A key behavioral measure in this task is the acceptance rate, which reflects participants' willingness to exert effort for reward. As expected, we replicated the classical pattern: acceptance rates decreased with increasing effort and increased with increasing reward (Fig. 1B). Interestingly, different components of behavior in this task have previously been linked to apathy-related deficits in multiple clinical populations[2,4–6,9–11]. We replicated this finding and confirmed that higher apathy scores were negatively correlated with acceptance rates in this task, specifically in high-reward trials (*e.g.*, 20-cent rewards at 80% of maximal effort; R = −0.29, p = .046; see Supplementary Fig. 2A). Supporting this, participants in the highest apathy quartile accepted fewer of the high-reward trials than those in the lowest quartile (0.68 ± 0.11 vs. 0.92 ± 0.04), reflecting a blunted behavioral drive to obtain high rewards. In addition, we analyzed a metric of effort exertion, defined as the excess contraction produced beyond the required level (*e.g.*, generating 35% of MVC in a trial with a 20% MVC target corresponds to 15% excess contraction), which provides an index of voluntary drive or over-commitment during effort execution. We found that higher apathy scores were significantly associated with lower excess force (*e.g.*, R = −0.39, p = 0.007 in 20% MVC/10 cents trials; see Supplementary Fig. 2.B), indicating reduced spontaneous effort investment. Together with the acceptance behavior results, these findings support the value of effort-based decision-making tasks for capturing inter-individual differences in motivational drive across both the decision and execution phases of goal-directed behavior.

As mentioned above, to further characterize the latent components of decision-making, we used computational modeling to estimate two key parameters: $\beta_{Effort}$, reflecting sensitivity to effort, and $\beta_{Reward}$, reflecting sensitivity to reward[6,9,11] (see Fig. 1B as well as Supplementary Fig. 3A–C for model selection and fitting). High $\beta_{Effort}$ and low $\beta_{Reward}$ values indicate increased effort cost valuation and reduced reward valuation — both features may be associated with reduced goal-directed behavior[6,9,11]. Nonetheless, neither $\beta_{Effort}$ nor $\beta_{Reward}$ significantly correlated with apathy scores in our sample. Bayesian regression analyses provided moderate evidence in favor of the null hypothesis (BF$_{01}$ = 2.97 for $\beta_{Effort}$; BF$_{01}$ = 2.24 for $\beta_{Reward}$; see Supplementary Fig. 3D). One possible explanation is that the effort and reward manipulations in this controlled laboratory task, although widely validated and commonly used to probe motivational processes, may not fully capture the complexity or ecological relevance of the cost-benefit trade-offs assessed with apathy questionnaires. As such, the latter relate more directly to complex, real-world goal-directed behaviors than to more simple decisions probed in laboratory tasks. Another explanation is that the model fitting process, by design, emphasizes latent constructs across trial types, averaging out trial-to-trial variability in acceptance rates that might reflect subtle motivational differences linked to apathy, particularly in conditions like high-reward trials (as shown in Supplementary Fig. 2A). This may limit the detection of significant correlations between apathy scores and model parameters. Still, model-derived parameters offer a robust framework to describe general effort and reward valuation processes, and are thus widely used in the field to capture inter-individual differences in these processes[2,4–6,9–11].

For structural connectivity, we acquired whole-brain DWI and performed streamline tractography[54,55] (Fig. 1C). We examined all tracts connecting the SMA and OFC to M1, both directly and through the BG structures identifiable using our atlas, all within the left hemisphere. The BG structures included the dorsal and ventral Caudate (d- and vCaudate), dorsolateral and ventromedial Putamen (dl- and vmPutamen), and the

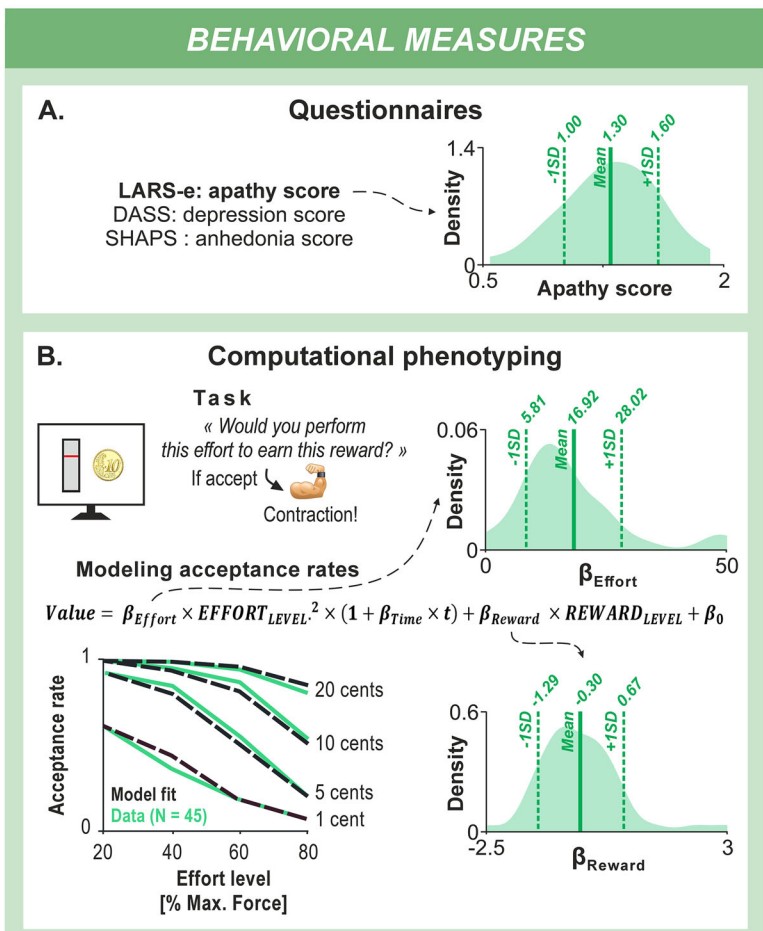

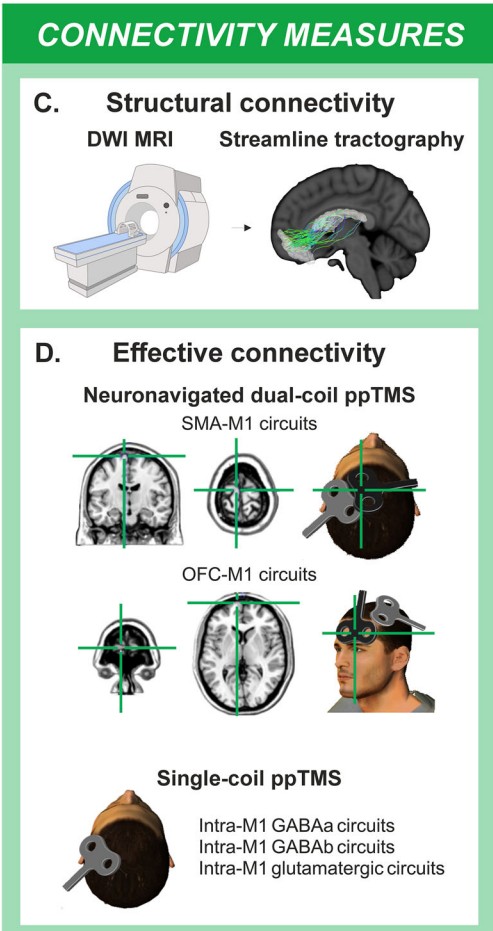

**Fig. 1 | Behavioral and connectivity measures. A** Neuropsychological questionnaires: We conducted a comprehensive neuropsychological assessment, including the extended version of the Lille Apathy Rating Scale (LARS-e) to quantify apathy scores, as well as the Depression Anxiety Stress Scales (DASS) and the Snaith-Hamilton Pleasure Scale (SHAPS) to quantify depression and anhedonia scores, respectively. The density distribution on the right illustrates the high inter-individual variability in apathy scores among our group of 45 healthy subjects. The goal of the present study was to better understand this variability in apathy levels in the general population. **B** Computational phenotyping: In the effort-based decision-making task (top left), participants decided whether to exert varying effort levels with their right biceps to earn different monetary rewards. Each trial displayed a required effort level (20–80% of MVC) via a vertical gauge, and a reward amount (1–20 euro cents). Participants had 5 seconds to accept or reject the offer using a keyboard. If they accepted, they attempted an isometric contraction monitored in real-time via EMG. Success required maintaining the target force for at least 2.7 seconds within a 4-second window. Accepted trials were rewarded if performance criteria were met; otherwise, no reward was given. The graph (bottom left) shows mean acceptance rates across effort levels for each reward amount in our sample of 45 subjects (green solid line), illustrating the expected pattern: acceptance increased with reward and decreased with effort (see also Supplementary Fig. 3; Source data for this plot are provided in Supplementary Data 1). Black dashed lines indicate the fit of our computational model. Parameters $\beta_{Effort}$ and $\beta_{Reward}$ were estimated from the modeling of acceptance rates using the equation presented on top of the graph (see Supplementary Fig. 3 for model comparison). Here again, the density distributions on the right depict the high inter-individual variability in $\beta_{Effort}$ and $\beta_{Reward}$ values among our group of 45 healthy subjects. **C** Structural connectivity: We measured DWI on the 45 subjects and performed streamline tractography on circuits connecting the SMA, OFC, BG, and M1 within the left hemisphere. The number of streamlines between pairs of structures was extracted for each subject, reflecting structural connectivity. **D** Effective connectivity: We used a dual-coil ppTMS protocol (top) to measure effective connectivity in circuits from the SMA and OFC to M1. We employed an individualized, neuronavigated dual-coil approach to investigate circuits from the SMA and the frontopolar portion of the OFC to M1 in the left hemisphere. Short (7 ms) and long (15 ms) inter-stimulation intervals were used to probe short-latency and long-latency circuits (not represented here), preferentially involving fronto-M1 and fronto-BG-M1 circuits, respectively[57,62]. We used a single-coil ppTMS approach (bottom) to examine intra-M1 GABAa, GABAb, and glutamatergic intra-M1 circuits. MEPs were recorded, and MEP ratios were calculated to assess effective connectivity (not represented here). The head cartoons were created using Poser software, version 6.

Nucleus Accumbens (NAcc). We also included the motor part of the thalamus to ensure the completeness of the circuits projecting to M1. We extracted the number of streamlines in each subject for 19 pairs of structures (detailed in Fig. 2). A higher number of streamlines between two structures in one subject compared to another, reflects a greater probability that the fibers linking these two regions are more numerous, occupy a larger volume, or are in better condition and thus indicate a greater structural connectivity.

For effective connectivity, we used ppTMS (Fig. 1D), which involved administering a test stimulation over the left M1, preceded by a conditioning stimulation (CS) in some trials[44,56,57,62]. We recorded the motor evoked

potentials (MEPs) induced by the test stimulation (TS) with EMG. We calculated a ratio of test MEPs for trials with both conditioning and test stimulations to trials with test stimulations only. A larger MEP ratio (> 1) indicates a stronger facilitatory drive from the pre-activated circuit to M1[44,56,57,62]. We used an individualized, neuronavigated dual-coil approach to investigate circuits from SMA and from the frontopolar portion of OFC to M1 in the left hemisphere. We used short (7 ms) and long (15 ms) inter-stimulation intervals to probe short-latency and long-latency circuits, involving preferentially fronto-M1 and fronto-BG-M1 circuits, respectively[57,62]. We also used a single-coil ppTMS approach to examine intra-M1 inhibitory and excitatory circuits. This technique relies on

**Article**

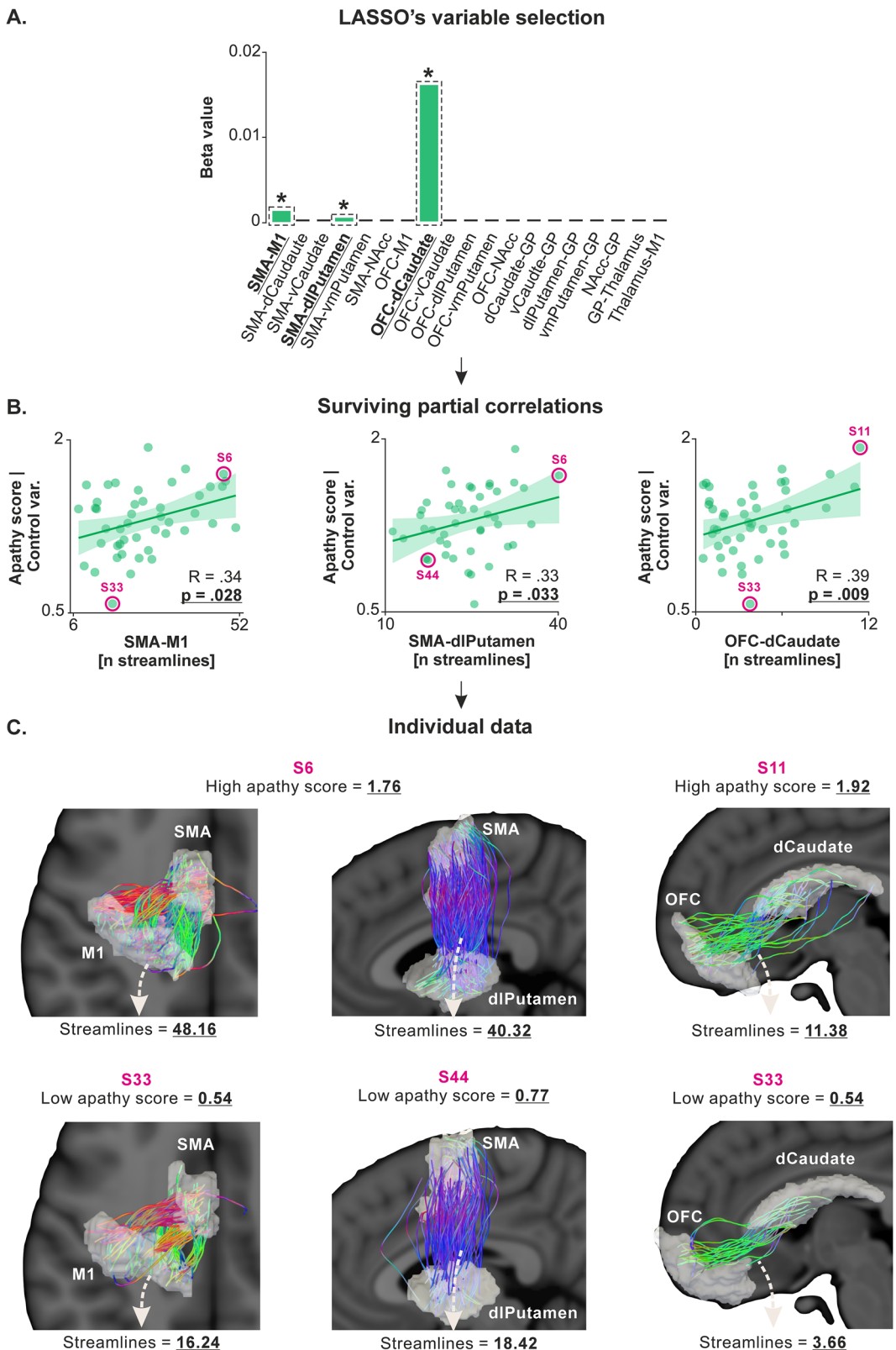

**A.** LASSO's variable selection

**B.** Surviving partial correlations

**C.** Individual data

systematic variations of the inter-stimulation interval (ISI) and the intensity of the CS to preferentially probe different circuits with one coil positioned over M1. Specifically, short ISIs (e.g., 3 ms) with subthreshold CS target fast, low-threshold $GABA_A$-mediated inhibition, as supported by pharmacological and electrophysiological studies (see ref. 56, for review). Then, longer ISIs (e.g., 100 ms) with suprathreshold CS engage slow, high-threshold GABAb-mediated inhibition, while intermediate ISIs (e.g., 10 ms) with subthreshold CS recruit glutamatergic neurons. The functional strength of these specific intra-M1 circuits is quantified by considering changes in the amplitude of MEPs elicited by the test stimulus (TS) when this TS is preceded by a CS, compared to when it is applied alone, providing a measure of intra-M1 effective connectivity[44,56].

**Fig. 2 | Apathy scores are linked to structural connectivity in SMA-M1, SMA-dlPutamen and OFC-dCaudate circuits. A** LASSO's variable selection. The bar graph shows the results of LASSO variable selection with structural connectivity data (*i.e.*, the number of streamlines for the 19 tracts considered) as independent variables and apathy scores as the dependent variable (N = 45 subjects). The regression identified three tracts (SMA-M1, SMA-dlPutamen, and OFC-dCaudate) with non-zero coefficients, suggesting a potential correlation with apathy scores. The green bars represent positive beta values, indicating a positive relationship with apathy scores. Bars highlighted by dashed black rectangles and a star indicate the independent variables that survived the ensuing partial correlation analysis. **B** Surviving partial correlations. Partial correlation analysis confirmed significant positive correlations between apathy scores and the number of streamlines in these three circuits (x-axes), even when controlling for potential confounding variables (*i.e.*, apathy scores adjusted for age, gender, depression, and anhedonia on the y-axes). Higher values on the x-axes reflect stronger structural connectivity in the considered circuit. Similarly, higher values on the y-axes reflect higher apathy scores. Source data for these plots are provided in Supplementary Data 2. **C** Individual data. Subjects with high apathy scores (top row, S6 and S11) and low apathy scores (bottom row, S33 and S44) were selected from the correlations presented in the middle panel, and their white matter for the three selected circuits were plotted. Note that the regions of interest have been inflated for the streamline analysis and are thus inflated in this visual representation as well. This representation highlights the strong heterogeneity in structural connectivity, even within a sample of healthy subjects, and its relationship with apathy scores.

Our study aimed to identify which independent variables quantifying structural connectivity (number of streamlines in 19 tracts) and effective connectivity (MEP ratios) are associated with three dependent variables: apathy scores, $\beta_{Effort}$ and $\beta_{Reward}$. We employed a conservative two-step statistical approach. First, we used Least Absolute Shrinkage and Selection Operator (LASSO) regression, a penalized least squares method that selects relevant independent variables (see also Supplementary Fig. 4 for a replication of the main findings using a less conservative Elastic Net regression). LASSO can yield some regression coefficients ($\beta$ coefficients) as zero. Only independent variables with non-zero regression coefficients are considered as associated with the dependent variable and thus selected. Second, we performed partial correlations between the dependent variables and the independent variables selected by the LASSO regression. This was done to control for multiple potentially confounding variables that may covary with apathy and effort-based decision-making processes (*i.e.*, age, gender, depression, and anhedonia), ensuring that the associations we identified between apathy scores, $\beta_{Effort}$, and $\beta_{Reward}$, on the one hand, and structural or effective connectivity, on the other, were truly specific to the dependent variable considered.

## Apathy scores are linked to connectivity in SMA-M1, SMA-BG and OFC-BG-M1 circuits

First, for structural connectivity, we performed a LASSO regression with the number of streamlines for the 19 tracts as independent variables and apathy scores as the dependent variable. The regression identified three tracts, namely SMA-M1, SMA-dlPutamen, and OFC-dCaudate, with non-zero coefficients, suggesting potential correlation with apathy scores (LASSO's $\beta$ coefficients = 0.001, 0.00005, and 0.016, respectively; Fig. 2A). Partial correlation analysis confirmed significant positive correlations between apathy scores and the number of streamlines in these circuits, even when controlling for the potentially confounding variables (R = 0.34, $p = 0.028$; R = 0.33, $p = 0.033$; and R = 0.39 $p = 0.009$, respectively; Fig. 2B), indicating that higher apathy scores are associated with stronger structural connectivity in these circuits. Interestingly, a second-level analysis of apathy subscores revealed that streamline count in the OFC-dCaudate tract was preferentially associated with the behavioral dimension of apathy (i.e., assessed through the action initiation subscore of the apathy scale; R = 0.31, $p = 0.015$; Supplementary Fig. 5A), but not with emotional or cognitive dimensions of apathy (R = −0.002, $p = 0.998$, and R = 0.13, $p = 0.405$, respectively), whereas connectivity in the SMA-dlPutamen tract covaried more specifically with the emotional component (R = 0.32, $p = 0.045$). These findings suggest that the observed associations with global apathy scores may reflect distinct contributions of fronto-striatal circuits to specific apathy dimensions.

As highlighted above, LASSO regression is a penalized least squares method that eliminates irrelevant independent variables by reducing their regression coefficients to zero. To ensure no false negatives in LASSO's selection, we performed additional correlations with BF computation on non-selected tracts. This involved all other SMA-BG and OFC-BG circuits. The analysis confirmed no significant positive correlations between apathy scores and the number of streamlines in these alternative circuits (*p*-values range = [0.437–0.872] and [0.199–0.826] for SMA-BG and OFC-BG circuits, respectively). The $BF_{01}$ for these correlations averaged 2.47 ± 0.05 ($BF_{01}$

range = [1.78–3.24]), indicating a higher likelihood of no correlation for these circuits. This analysis confirms the absence of false negatives in LASSO's selection and emphasizes the circuit-specificity of the significant correlations.

Second, for effective connectivity, we applied LASSO regression with all MEP ratios as independent variables and apathy scores as the dependent variable. This included ratios for short- and long-latency SMA- and OFC-M1 circuits, as well as intra-M1 GABAa, GABAb, and glutamatergic circuits. The regression revealed two circuits (short- and long-latency OFC-M1) with non-zero coefficients (LASSO's $\beta$ coefficients = −0.23 and 0.61, respectively; Fig. 3A). Importantly, partial correlation analyses confirmed a significant positive correlation for the long-latency OFC-M1 circuit (R = 0.38, $p = 0.014$; Fig. 3B), indicating that higher apathy scores are associated with a stronger facilitatory influence of OFC on M1 (*i.e.*, MEP ratios > 1) in this long-latency circuit. This effect aligns with the positive relationship observed between apathy scores and OFC–dCaudate structural connectivity (Fig. 2), making it one of the most robust findings of the study, as it emerges across two independent modalities of connectivity quantification. Besides, apathy scores were not associated with connectivity in the short-latency OFC-M1 circuit (R = −0.09, $p = 0.571$; Supplementary Fig. 6), showing that this effect was specific to the long-latency circuit potentially involving the BG. Interestingly, the significant positive correlation for the long-latency OFC-M1 circuit was specific to the behavioral dimension of apathy (R = 0.37, $p = 0.015$), but not to the emotional or cognitive components (R = 0.22, $p = 0.152$, and R = 0.19, $p = 0.222$, respectively; Supplementary Fig. 5B). This mirrors the similar association found for OFC-dCaudate structural connectivity, suggesting that disruptions in OFC-NG-M1 circuits may selectively contribute to reduced action initiation.

Hence, our findings establish a link between apathy scores and connectivity in different SMA and OFC circuits part of which connect to M1. Notably, apathy scores showed a positive correlation with both structural and effective connectivity in OFC-BG-M1 circuits. However, the association observed with structural connectivity in SMA-M1 and SMA-dlPutamen circuits did not extend to effective connectivity. All observed correlations were positive, indicating that higher levels of apathy are associated with a stronger structural connectivity in these circuits and a stronger facilitatory influence from OFC to M1. This aligns with previous reports linking high subclinical apathy levels to increased activation[17] and hyperconnectivity[63] in frontal structures.

## Effort valuation is linked to structural connectivity in SMA-M1 and SMA-BG circuits

We aimed to characterize the brain circuits associated with increasing $\beta_{Effort}$ values. High $\beta_{Effort}$ values are of particular interest because they reflect a stronger reduction in acceptance rates as effort levels increase. In other words, high $\beta_{Effort}$ values indicate a greater sensitivity to effort costs, which is linked to reduced engagement in effortful tasks[6,11,17].

The relationship between connectivity and effort valuation was analyzed using the same statistical procedure as for apathy scores. First, we performed a LASSO regression, with the number of streamlines for the 19 tracts as independent variables and $\beta_{Effort}$ as the dependent variable. The regression identified seven tracts with non-zero coefficients: SMA-M1 (LASSO's $\beta = 0.21$), SMA-dlPutamen ($\beta = -0.02$), SMA-vmPutamen ($\beta = -0.31$),

## A. LASSO's variable selection

## B. Surviving partial correlation

## C. Individual data

**Fig. 3 | Apathy scores are linked to effective connectivity in the long-latency OFC-M1 circuit. A** LASSO's variable selection. The bar graph shows the results of LASSO variable selection, with effective connectivity data (*i.e.*, the MEP ratios) as independent variables and apathy scores as the dependent variable (N = 45 subjects). The regression identified two circuits (short- and long-latency OFC-M1 circuits) with non-zero coefficients, suggesting a potential correlation with apathy scores. The red and green bars represent negative and positive beta values, indicating negative and positive relationships with apathy scores. The bar highlighted by a dashed black rectangle and a star indicates the independent variable that survived the ensuing partial correlation analysis. **B** Surviving partial correlation. Partial correlation analysis confirmed significant positive correlations between apathy scores and MEP ratios for the long-latency OFC-M1 circuits (x-axes), even when controlling for potential confounding variables (*i.e.*, apathy scores adjusted for age, gender, depression, and anhedonia on the y-axes). However, this was not the case for the short-latency OFC-M1 circuit (R = −0.09, p = 0.571; Supplementary Fig. 6). Higher values on the y-axes reflect higher apathy scores. Similarly, higher values on the x-axis (MEP ratios > 1) reflect stronger facilitatory influence of OFC on M1. Notably, the distribution of MEP ratios is centered near 1, with participants displaying both < 1 and > 1 values, indicating that OFC can exert either an inhibitory or facilitatory influence on M1 at rest. This inter-individual variability is consistent with prior ppTMS studies of long-range OFC-M1 connectivity, which have also reported mixed directionality across individuals[57]. Source data for these plots are provided in Supplementary Data 3. **C** Individual data. Subjects with high apathy scores (top row, S30) and low apathy scores (bottom row, S9) were selected from the correlations presented in the middle panel, and their MEPs were plotted. The plots represent the average MEPs time-locked to the TMS pulse; shaded errors represent standard errors. Test MEPs were obtained with a single test stimulation over M1 while conditioned MEPs were obtained by first applying a conditioning stimulation on OFC, 15 ms before the test stimulation. S30 (top row) presents a high apathy score and a larger conditioned MEP amplitude relative to test MEP amplitude. Conversely, S9 (bottom row) presents a low apathy score and a tendency for a lower conditioned MEP amplitude relative to test MEP amplitude. These plots highlight the strong heterogeneity in effective connectivity between OFC and M1, even within a sample of healthy subjects, and its relationship with apathy scores.

streamlines in SMA-M1 circuit was positive while the correlations with SMA-NAcc and NAcc-GP circuits were negative, indicating that higher levels of effort valuation are associated with stronger cortico-cortical connectivity and weaker cortico-subcortical connectivity in SMA-related circuits. Other partial correlations were not significant and are presented in Supplementary Fig. 7 (*p*-values range = [0.198–0.784]).

Second, we applied LASSO regression with MEP ratios as independent variables and $\beta_{Effort}$ as the dependent variable. The regression yielded zero coefficients for all circuits (*i.e.*, all β = 0), signifying a lack of association between $\beta_{Effort}$ and effective connectivity in the investigated circuits. All graphs depicting the results of the LASSO regression as well as the absence of significant partial correlation between all effective connectivity data and $\beta_{Effort}$ are provided in Supplementary Fig. 8 (*p*-values range = [0.385–0.887]).

Hence, our findings establish a link between effort valuation and structural connectivity in different SMA-related – and not OFC-related – circuits, in line with the well-known role of SMA in effort valuation[29,30]. However, as for apathy scores, the associations observed with structural connectivity in SMA-related circuits did not extend to effective connectivity alterations, in line with the idea that structural and effective connectivity measures provide complementary insights into neural circuits and do not necessarily co-vary with one another. Further, the correlation was positive for the SMA-M1 circuit and negative for SMA-NAcc and NAcc-GP circuits, indicating that effort valuation might be associated with different alterations in distinct fronto-motor circuits.

### Reward valuation is linked to structural connectivity in the NAcc-GP circuit and to effective connectivity in intra-M1 GABAa and GABAb circuits

We aimed to characterize the brain circuits associated with decreasing $\beta_{Reward}$ values. Low $\beta_{Reward}$ values are of particular interest because they

SMA-NAcc (β = −0.25), vCaudate-GP (β = −0.04), NAcc-GP (β = −0.11) and GP-Thalamus (β = 0.02; Fig. 4A). Partial correlation analyses confirmed significant correlations between $\beta_{Effort}$ and the number of streamlines in three of these seven tracts, namely SMA-M1 (R = 0.42, *p* = 0.006), SMA-NAcc (R = −0.32, *p* = 0.042) and NAcc-GP circuits (R = −0.41, *p* = 0.007; Fig. 4B). As indicated by the sign of the R-values, the correlation with the number of

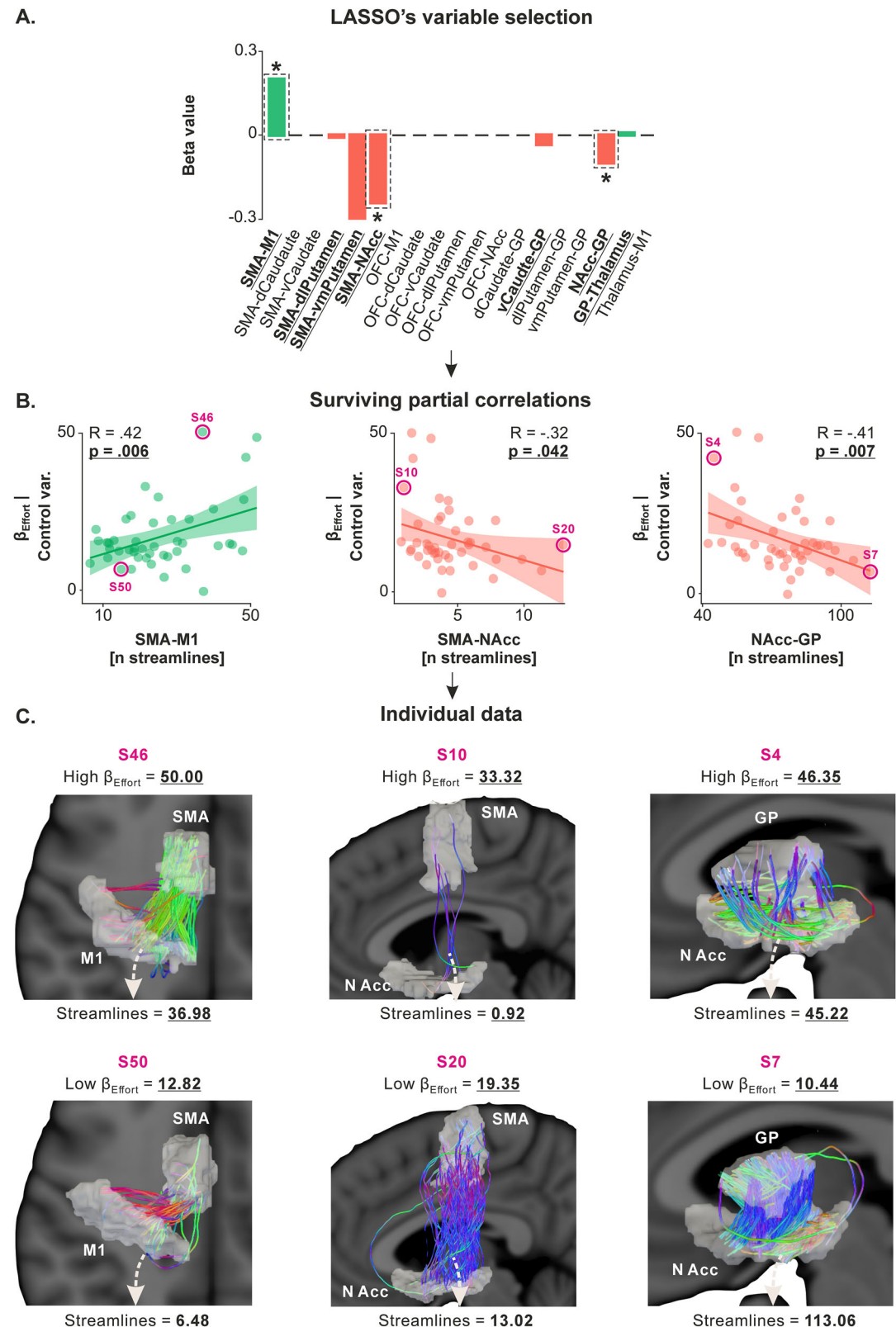

**A.** LASSO's variable selection

**B.** Surviving partial correlations

**C.** Individual data

reflect a lower increase in acceptance rates as reward levels increase. In other words, low $\beta_{Reward}$ values indicate a lower sensitivity to rewards, which is linked to reduced engagement in effortful tasks[6,9,10].

The relationship between connectivity and reward valuation was analyzed using the same statistical procedure as for the other dependent variables. First, we performed the LASSO regression, with the number of

streamlines for the 19 tracts as independent variables and $\beta_{Reward}$ as the dependent variable, and identified one tract with a non-zero coefficient (NAcc-GP, LASSO's β coefficient = −0.012; Fig. 5A). The partial correlation analysis confirmed the presence of a significant negative correlation between $\beta_{Reward}$ and the number of streamlines in the NAcc-GP circuits (R = −0.42, $p$ = 0.006; Fig. 5B), consistent with the well-identified role of NAcc in value

**Fig. 4 | $\beta_{\text{Effort}}$ is linked to structural connectivity in SMA-M1, SMA-NAcc and NAcc-GP circuits. A** LASSO's variable selection. The bar graph shows the results of LASSO variable selection, with structural connectivity data (*i.e.*, the number of streamlines for the 19 tracts considered) as independent variables and $\beta_{\text{Effort}}$ as the dependent variable ($N = 45$ subjects). The regression identified seven tracts (SMA-M1, SMA-dlPutamen, SMA-vmPutamen, SMA-NAcc, vCaudate-GP, NAcc-GP and GP-Thalamus) with non-zero coefficients, suggesting a potential correlation with $\beta_{\text{Effort}}$. The red and green bars represent negative and positive beta values, indicating negative and positive relationships with $\beta_{\text{Effort}}$. Bars highlighted by dashed black rectangles and a star indicate the independent variables that survived the ensuing partial correlation analysis. **B** Surviving partial correlations. Partial correlation analysis confirmed significant correlations between $\beta_{\text{Effort}}$ and the number of streamlines in three of the seven circuits (x-axes), even when controlling for potential confounding variables (*i.e.*, $\beta_{\text{Effort}}$ adjusted for age, gender, depression, and anhedonia on the y-axes). Higher values on the x-axes reflect stronger structural connectivity in the considered circuit. Similarly, higher values on the y-axes reflect higher $\beta_{\text{Effort}}$, indicating stronger valuation of efforts in the effort-based decision-making task. Other partial correlations were not significant and are presented in Supplementary Fig. 7 (*p*-values range = [0.198 –0.784]). Source data for these plots are provided in Supplementary Data 4. **C** Individual data. Subjects with high $\beta_{\text{Effort}}$ (top row, S46, S10 and S4) and low $\beta_{\text{Effort}}$ (bottom row, S50, S20 and S7) were selected from the correlations presented in the middle panel, and their white matter for the three selected circuits were plotted. Note that the regions of interest have been inflated for the streamline analysis and are thus inflated in this visual representation as well. This representation highlights the strong heterogeneity in structural connectivity, even within a sample of healthy subjects, and its relationship with $\beta_{\text{Effort}}$.

computation[64,65]. As such, the negative correlation indicates that lower $\beta_{\text{Reward}}$ parameters – *i.e.*, lower levels of reward valuation – are associated with stronger structural connectivity in this circuit.

We then applied the LASSO regression with all MEP ratios as independent variables and $\beta_{\text{Reward}}$ as the dependent variable and identified five circuits with non-zero coefficients (Fig. 6A): intra-M1 GABAa (LASSO's $\beta = -0.015$), intra-M1 GABAb ($\beta = -0.006$), intra-M1 glutamatergic ($\beta = 0.002$), short-latency OFC-M1 ($\beta = 0.005$) and long-latency OFC-M1 circuits ($\beta = -0.0004$). Partial correlation analyses confirmed the presence of a significant correlation between $\beta_{\text{Reward}}$ and two of the five circuits, namely intra-M1 GABAa ($R = -0.50$, $p = 0.0007$) and intra-M1 GABAb circuits ($R = -0.31$, $p = 0.047$; Fig. 6B), two circuits that are known to be modulated during reward-based tasks[66,67]. As indicated by the sign of the R-values, the correlations between $\beta_{\text{Reward}}$ and effective connectivity in intra-M1 GABAa and GABAb circuits was negative. The negative correlations indicate that participants with lower $\beta_{\text{Reward}}$, reflecting lower sensitivity to rewards (*i.e.*, those at the bottom of the y-axis in Fig. 6B), had reduced effective connectivity in intra-M1 GABAa and GABAb circuits. Reduced effective connectivity suggests less inhibitory influence from these GABAergic circuits on corticospinal neurons (*i.e.*, a reduction in the suppressive effect of GABAa and GABAb circuits, seen on the right side of the x-axis in Fig. 6B). In simpler terms, individuals who placed a lower value on rewards exhibited less GABA-mediated inhibition within M1. The other partial correlations were not significant and are presented in Supplementary Fig. 9 (*p*-values range = [0.101–0.907]).

Hence, our findings establish a link between reward valuation and structural connectivity in an isolated tract of the BG circuitry – *i.e.*, the NAcc-GP tract. Interestingly, this tract's connectivity is also tied to effort valuation, supporting the NAcc's role in integrating reward and effort values to compute the net value of actions[13,37,68]. We also found a negative link between reward valuation and effective connectivity in intra-M1 GABAa and GABAb circuits. This means lower reward valuation is linked to less inhibition of corticospinal neurons by these circuits, aligning with literature showing GABAergic circuits' role in reward-based behaviors[66,67]. A summary of the correlations we uncovered is presented in Fig. 7.

### Apathy scores, effort valuation and reward valuation are not linked to M1 net corticospinal output

The data presented above show correlations between apathy scores, $\beta_{\text{Effort}}$ and $\beta_{\text{Reward}}$, on the one hand, and both structural and effective connectivity, on the other, including circuits projecting to and within M1. Given this, it is possible that variability in connectivity towards M1 could lead to differences in M1's net neural output across apathy scores, $\beta_{\text{Effort}}$ and $\beta_{\text{Reward}}$ values. If that were the case, M1 net corticospinal output would be, by itself, linked to apathy scores, $\beta_{\text{Effort}}$, and $\beta_{\text{Reward}}$.

To test this idea, we performed partial correlations with apathy scores, $\beta_{\text{Effort}}$, and $\beta_{\text{Reward}}$ as dependent variables and test MEP amplitudes (obtained with a single test stimulation over M1), a proxy for M1 net output, as the independent variable. Control variables included age, gender, depression, and anhedonia. Partial correlations showed no significant link between apathy scores ($R = -0.11$, $p = 0.468$), $\beta_{\text{Effort}}$ ($R = 0.05$, $p = 0.737$),

$\beta_{\text{Reward}}$ ($R = -0.12$, $p = 0.446$) and test MEP amplitudes (Supplementary Fig. 10). BF computation indicated a higher likelihood of no correlation ($\text{BF}_{01}$ averaged $2.59 \pm 0.34$, range = [2.12–3.25]).

Thus, despite correlations between apathy scores, effort valuation, reward valuation, and connectivity in circuits projecting to M1, these dependent variables are not linked to M1 net corticospinal output alone. This analysis is also an important methodological control as it shows that the relationships between apathy scores, $\beta_{\text{Effort}}$, $\beta_{\text{Reward}}$, and MEP ratios are not due to associations with the test MEP amplitudes exploited to compute these ratios.

## Discussion

This study probed the relationship between apathy and effort-based decision-making, and connectivity in circuits linking key fronto-BG structures to M1. Our data showed that different variables characterizing apathy and effort-based decision-making processes – namely apathy scores, $\beta_{\text{Effort}}$, and $\beta_{\text{Reward}}$ – were linked to distinct brain circuits, some of which project to M1. First, apathy scores were linked to connectivity in both SMA- and OFC-originating circuits. Specifically, our tractography data showed stronger structural connectivity in SMA-M1, SMA-BG, and OFC-BG circuits as apathy scores increase. Additionally, ppTMS data highlighted a positive correlation between facilitatory long-latency OFC-M1 effective connectivity and apathy scores, likely mediated by an OFC-BG-M1 circuit. Second, effort valuation was linked exclusively to SMA-related circuits. Individuals with higher $\beta_{\text{Effort}}$ showed stronger structural connectivity in SMA-M1 and SMA-BG circuits, supporting the known role of SMA in effort valuation. Third, reward valuation was associated with structural connectivity in the NAcc-GP BG tract, with lower $\beta_{\text{Reward}}$ (indicative of a reduced engagement for increasing rewards) observed in subjects showing stronger connectivity. Finally, lower $\beta_{\text{Reward}}$ was also associated with lower effective connectivity in intra-M1 GABAergic circuits.

Interestingly, much of the relationships we uncovered suggest that reduced engagement in effort (*i.e.*, higher apathy scores, higher $\beta_{\text{Effort}}$, and lower $\beta_{\text{Reward}}$) is linked to stronger structural and effective connectivity. This applies across several circuits, including SMA- and OFC-originating circuits (higher apathy), the SMA-M1 circuit (higher effort valuation), and the NAcc-GP circuit (lower reward valuation). At first glance, these findings may seem at odds with part of the literature showing links between both subclinical and clinical apathy and brain abnormalities such as grey matter atrophy[35,69–71], hypoactivation[72], hypometabolism[73], white matter lesions[17,69], and reduced functional connectivity[26,74,75]. However, as mentioned in the Introduction, prior studies have reported mixed findings, with both hypo- and hyper-connectivity patterns observed in relation to apathy, suggesting that either reduced or excessive connectivity may reflect maladaptive network dynamics contributing to motivational impairments.

In fact, based on the literature, such findings of hyperconnectivity may be interpreted in two ways. First, increased connectivity in these circuits might be detrimental to effort-based decision-making. High functional or effective connectivity at rest may reduce the brain's ability to adjust dynamically to different effort and reward magnitudes during decision-making. This kind of hyperconnectivity is seen in various disorders where different

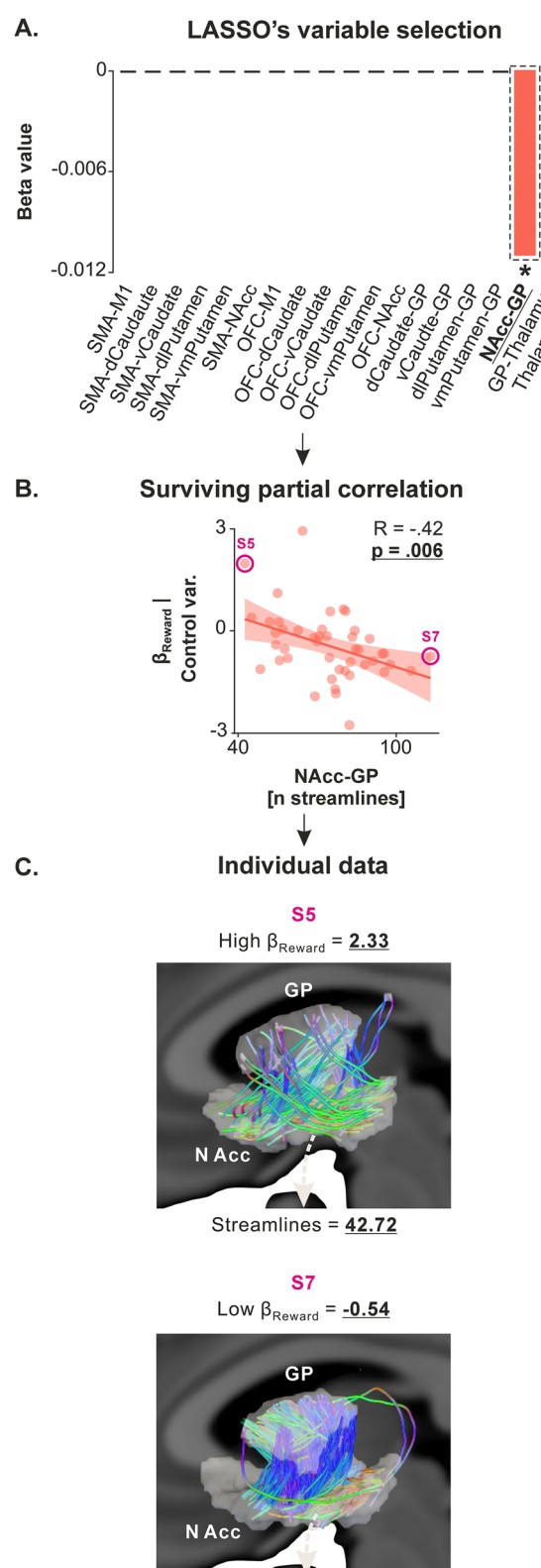

**A. LASSO's variable selection**

**B. Surviving partial correlation**

**C. Individual data**

S5
High β_Reward = **2.33**

GP

N Acc

Streamlines = **42.72**

S7
Low β_Reward = **-0.54**

GP

N Acc

Streamlines = **113.06**

**Fig. 5 | β_Reward is linked to structural connectivity in the NAcc-GP circuit.**
**A** LASSO's variable selection. The bar graph shows the results of LASSO variable selection, with structural connectivity data (*i.e.*, the number of streamlines for the 19 tracts considered) as independent variables and β_Reward as the dependent variable ($N = 45$ subjects). The regression identified one tract (NAcc-GP) with a non-zero coefficient, suggesting a potential correlation with β_Reward. The red bar represents a negative beta value, indicating a negative relationship with β_Reward. This bar is highlighted by a dashed black rectangle and a star, indicating that it survived the ensuing partial correlation analysis. **B** Surviving partial correlation. Partial correlation analysis confirmed the presence of a significant negative correlation between β_Reward and the number of streamlines in the NAcc-GP circuits (x-axes), even when controlling for potential confounding variables (*i.e.*, β_Reward adjusted for age, gender, depression, and anhedonia on the y-axes). Higher values on the x-axes reflect stronger structural connectivity in the NAcc-GP circuit. Further, lower values on the y-axes reflect lower β_Reward, indicating lower valuation of rewards in the effort-based decision-making task. Source data for these plots are provided in Supplementary Data 5. **C** Individual data. Subjects with high β_Reward (top row, S5) and low β_Reward (bottom row, S7) were selected from the correlations presented in the middle panel, and their white matter for the NAcc-GP circuit were plotted. Note that the regions of interest have been inflated for the streamline analysis and are thus inflated in this visual representation as well. This representation highlights the strong heterogeneity in structural connectivity, even within a sample of healthy subjects, and its relationship with β_Reward.

circuits[81–83]. On the other hand, heightened connectivity might help rather than hinder. It could enhance neural communication during effort-based decisions, helping individuals with subclinical apathy to avoid tipping into clinical apathy. For example, in traumatic brain injury patients, those who show high subclinical apathy levels have hyperconnectivity in frontal circuits, while those who develop clinical apathy show reduced connectivity in these same circuits[63]. This suggest that hyperconnectivity could reflect an initial stage of compensation in specific fronto-motor circuits in individuals with mild apathy that is ultimately insufficient as the pathology progresses[63]. If this is correct, the relationship between connectivity and apathy might follow a non-linear path in certain circuits. Starting with normal connectivity at low apathy levels, it could progress to hyperconnectivity at higher subclinical levels, and eventually to hypoconnectivity in more severe, clinical cases. Understanding this dynamic could pave the way for developing diagnostic biomarkers that track different stages of apathy across individuals, and design stage-dependent neuromodulation approaches.

Among the positive correlations we found, high apathy scores were linked to hyperconnectivity in several circuits originating from both SMA and OFC, partially projecting to M1. This finding is significant for two reasons. First, it contrasts with the more specialized circuits involved in effort valuation (SMA circuits) and reward valuation (NAcc and GABAergic circuits). This supports the idea that apathy scores may capture broader processes driving motivated behavior[10], engaging multiple brain circuits with different roles. In contrast, effort and reward valuation, as captured by computational models, may engage more specific circuits. Second, it sheds light on M1, a region often overlooked in studies of effort-based decision-making and apathy. This oversight likely stems from the influence of serial architecture models, which suggest that motor regions only come into play after a decision has been made, to execute movements[84,85]. However, research in rodents, non-human primates, and humans has consistently shown that M1 activity changes during decision-making, especially in response to effort and reward[45,47,50]. This supports the idea that decision-making relies on a distributed consensus across brain regions[42,44], with connectivity modulations within fronto-BG-M1 recurrent loops playing a more active role than previously thought[53,86]. On a practical level, the association of fronto-motor circuits including M1 with apathy opens new translational opportunities to modulate and probe these circuits using multifocal non-invasive brain stimulation techniques such as paired-associative TMS[87–89] or bifocal transcranial alternating current stimulation approaches[90]. Although our findings are correlational and based on structural and resting-state measures, they align with the idea that connectivity in fronto-BG-M1 circuits is crucial for multiple dimensions of apathetic behavior.

aspects of motivated behavior are affected. For instance, schizophrenia is associated with both negative symptoms, such as avolition and asociality, and increased functional connectivity[76]. Depression is linked to both anhedonia and an expansion of brain networks[77] as well as hyperconnectivity in specific brain networks[78–80]. In Parkinson's disease, patients exhibit reduced movement vigor and excessive beta band connectivity in fronto-BG

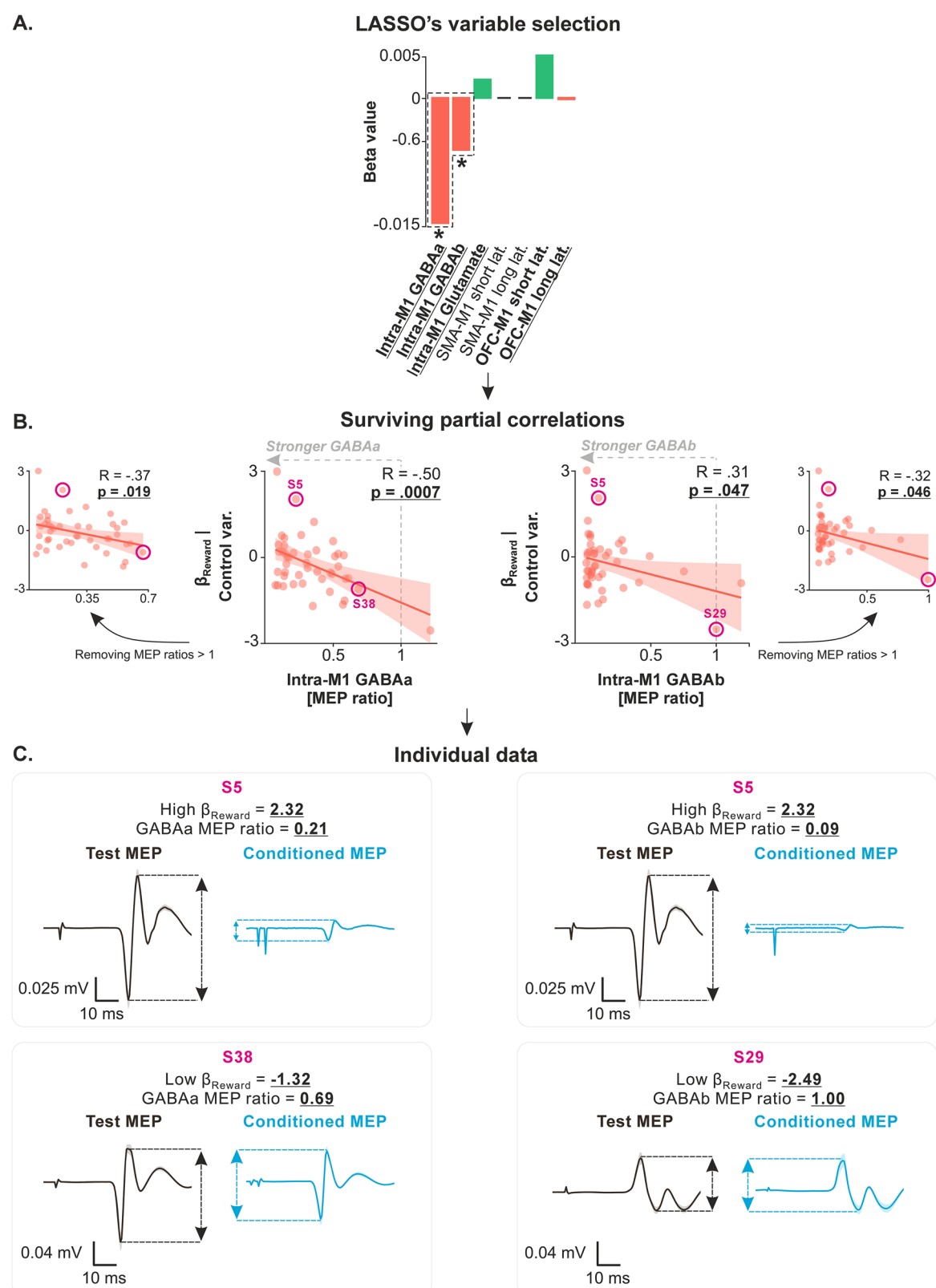

Higher effort valuation was also linked to stronger structural connectivity between SMA and M1, which was also associated with higher apathy scores. This correlation is interesting because SMA activity has been shown to scale with the intensity of muscle contraction[23,29], and disruption of the SMA alters effort perception[30,91]. This supports the idea that the SMA plays a key role in encoding effort costs[30,91]. Additionally, research on motor imagery suggests that SMA suppresses M1 activity when imagining movements without actually performing them[92]. One possibility is that stronger SMA-M1 connectivity enhances the SMA's ability to suppress motor activity when mentally simulating actions involving high efforts. Another possibility is that the increased connectivity we found stems from projections running in the opposite direction, from M1 to SMA, given that

**Fig. 6 | $\beta_{Reward}$ is linked to effective connectivity in intra-M1 GABAa and GABAb circuits. A** LASSO's variable selection. The bar graph shows the results of LASSO variable selection, with effective connectivity data (*i.e.*, the MEP ratios) as independent variables and $\beta_{Reward}$ as the dependent variable ($N = 45$ subjects). The regression identified five circuits tracts with non-zero coefficients, suggesting a potential correlation with $\beta_{Reward}$. The red and green bars represent negative and positive beta values, indicating negative and positive relationships with apathy scores. Bars highlighted by dashed black rectangles and a star indicate the independent variables that survived the ensuing partial correlation analysis. **B** Surviving partial correlations. Partial correlation analysis confirmed significant negative correlations between $\beta_{Reward}$ and MEP ratios for two of the five circuits: intra-M1 GABAa (R = −0.50, $p = 0.0007$) and intra-M1 GABAb circuits (x-axes), even when controlling for potential confounding variables (*i.e.*, $\beta_{Reward}$ adjusted for age, gender, depression, and anhedonia on the y-axes). Other partial correlations were not significant and are presented in Supplementary information 9 for the sake of transparency ($p$-values range = [0.101–0.907]). Lower values on the x-axis (MEP ratios <1 and tending toward 0) reflect stronger suppressive influence of the GABAa and GABAb circuits on corticospinal neurons. Additionally, lower values on the y-axes reflect lower $\beta_{Reward}$ and thus a lower valuation of rewards in the effort-based decision-making task. Source data for these plots are provided in Supplementary Data 6. **C** Individual data. Subjects with high $\beta_{Reward}$ (top row, S5) and low $\beta_{Reward}$ (bottom row, S38 and S29) were selected from the correlations presented in the middle panel, and their MEPs were plotted. The plots represent the average MEPs time-locked to the TMS pulse; shaded errors represent standard errors. Test MEPs were obtained with a single test stimulation over M1 while conditioned MEPs were obtained by first applying a conditioning stimulation on M1 before the test stimulation. S30 (top row) presents a high $\beta_{Reward}$ and a strong suppression of conditioned MEP amplitude relative to test MEP amplitude both for GABAa and GABAb circuits, indicating a strong effective, suppressive influence of these circuits on corticospinal neurons. Conversely, S38 and S29 (bottom row) presents a low $\beta_{Reward}$ and almost no suppression of conditioned MEP amplitudes relative to test MEP amplitudes, indicating a weak suppressive influence of GABA circuits on corticospinal neurons. These plots highlight the strong heterogeneity in effective connectivity in intra-M1 GABAa and GABAb circuits, even within a sample of healthy subjects, and its relationship with $\beta_{Reward}$.

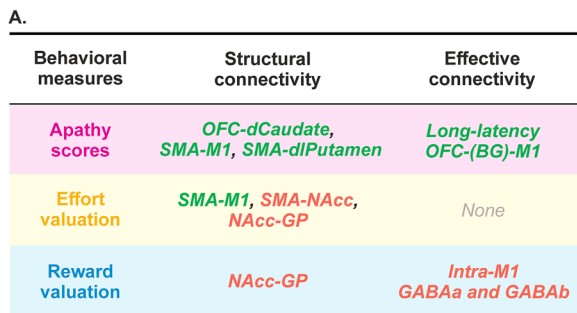
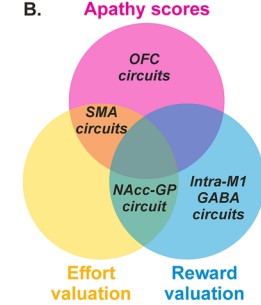

**Fig. 7 | Summary of the main correlations uncovered in the present study. A** Detailed correlations. Green and red colors are used to highlight positive and negative correlations, respectively. Apathy scores are linked to structural connectivity in the SMA-M1, SMA-dlPutamen, and OFC-dCaudate circuits, as well as to long-latency OFC-M1 effective connectivity. Effort valuation is associated with structural connectivity in the SMA-M1, SMA-NAcc, and NAcc-GP circuits. Reward valuation is linked to structural connectivity in the NAcc-GP circuit and effective connectivity in intra-M1 GABAergic circuits. Notably, both OFC–BG structural connectivity and OFC–M1 long-latency effective connectivity showed positive correlations with apathy scores (and, more specifically, behavioral apathy; see Supplementary Fig. 5). Hence, this relationship was replicated across two independent modalities (DWI and dual-coil ppTMS). In addition, SMA–M1 structural connectivity was associated with both apathy scores and effort valuation, which is highly consistent with the SMA's established role in effort processing. **B** Overlapping and specific circuits. This Venn diagram provides an overview of the circuits shared and specific to apathy scores, effort valuation, and reward valuation.

these regions are densely, reciprocally connected[93–95]. Our data showed no link between effort valuation or apathy scores and short-latency effective connectivity from SMA to M1, suggesting that structural connectivity changes in SMA-M1 projections alone may not explain the observed effects. Stronger M1-SMA connectivity could allow M1 to communicate more efficiently the body's state to SMA, improving effort estimation during embodied decisions. Further research is needed to clarify the specific roles of SMA-M1 and M1-SMA circuits in effort valuation, and their relationship to apathy.

Our data also reveal that higher effort valuation is linked to weaker structural connectivity in the SMA-NAcc and NAcc-GP circuits. Interestingly, weaker NAcc-GP connectivity is also tied to higher reward valuation. In other words, individuals with weaker NAcc-GP connectivity are more sensitive to changes in both effort and reward. They reduce their acceptance rates more sharply as effort increases (higher $\beta_{Effort}$) and increase their acceptance rates more strongly with increasing rewards (higher $\beta_{Reward}$). This fits with the known role of the NAcc in computing the net value of actions, weighing effort and reward[37,68]. Further, the net impact of the NAcc-GP circuit on motor cortical output is mostly suppressive[96]. Indeed, the NAcc primarily projects through inhibitory medium spiny neurons to the pallidum[96–98], which in turn inhibits the suppressive influence of ventral mesencephalon structures on thalamo-cortical circuits via the so-called indirect pathway of the BG[96]. One possibility is that the weaker structural connectivity we found reflects fewer or altered axons of medium spiny neurons projecting from the NAcc to the pallidum. This could reduce their

suppressive effect on motor output, making individuals more sensitive to changes in both effort and reward levels.

Finally, lower reward valuation was also linked to weaker GABAa and GABAb effective connectivity within M1. This effect was specific to GABAergic circuits and did not concern glutamatergic ones. Importantly, a similar finding is observed in people with major depression, who present lower plasma GABA levels compared to controls, indicating reduced inhibitory function in individuals with impaired motivation[99]. Further, in task settings, GABAergic neurons in M1 adjust their activity in response to reward cues and delivery[66,67], acting as a "brake" on motor output that is released when rewards are available. These findings, along with our data, suggest that GABAergic circuits may play a role in regulating reward-driven behavior down to the final motor neurons of the brain. Specifically, the present study suggests a possible compensatory mechanism: individuals with higher reward valuation may develop more efficient inhibitory circuits to corticospinal neurons at rest, preventing triggering any action when high rewards are at stake. In contrast, those less sensitive to rewards may need less GABA-driven suppression of the motor system.

While our study focused on specific fronto-BG circuits projecting to M1, apathy is widely recognized as a network-level disorder. Beyond fronto-BG circuits, structural and functional MRI studies have shown alterations in large-scale networks, including the default mode and salience networks[100–102]. Our targeted approach was designed to test mechanistic hypotheses about the contribution of motor circuits to apathy, as well as to effort and reward valuation. Nonetheless, future studies incorporating

whole-brain structural and/or functional connectivity analyses could complement our work and help clarify how these motor pathways interact with broader large-scale networks

Overall, this study adds a missing piece to the puzzle of the apathy network. It reveals clear links between apathy levels and effort-based decision-making processes in the general population on the one hand, and specific fronto-BG circuits on the other, including some directly connected to the brain's output neurons in M1. These results also underscore the value of combining a detailed assessment of motivated behavior, including well-established computational approaches, with multimodal connectivity measures to better understand the neural mechanisms underlying apathy.

## Materials and methods
### Experimental model and subject details
The study initially involved 50 healthy human subjects recruited from the Research Participant Pool at the Institute of Neuroscience of UCLouvain. However, the experimental design consisted of two sessions and 3 out of the 50 subjects did not attend the second session. Additionally, tractography analysis could not be performed on 2 subjects due to corrupted MRI data. Consequently, the results presented in this article focus on the data of the 45 remaining subjects (25.1 ± 0.8 years old, 31 females, 14 males).

All subjects were right-handed according to the Edinburgh Questionnaire (Oldfield, 1971). None of the participants had any neurological disorder or history of psychiatric illness or drug or alcohol abuse, or were on any drug treatments that could influence performance. Participants were financially compensated for their participation and earned additional money depending on their performance on the task (see below). The protocol was approved by the institutional review board of the Université catholique de Louvain, Brussels, Belgium, and required written informed consent.

### Method details
The study consisted of two sessions, conducted with a minimum interval of 24 hours and a maximum interval of one week. The first session involved MRI data acquisitions and was performed at the Saint-Luc University Hospitals (Brussels, Belgium). The second session involved TMS data acquisition, behavioral data acquisition on the effort-based decision-making task, and the administration of a battery of neuropsychological questionnaires (i.e., systematically realized in this order). It was performed at the Institute of Neuroscience of UCLouvain in Brussels. Below, we provide specific details for each of these acquisitions.

**Magnetic resonance imaging**. A 3D T1-weighted image was acquired for each participant on a 3T GE SIGNA™ Premier scanner (GE Healthcare, Chicago, IL) with the following parameters: Echo Time (TE) = 2.96 ms, Repetition Time (TR) = 2238.93 ms, Inversion Time (TI) = 900 ms, 170 slices, slice thickness: 1 mm, in-plane Field Of View (FOV): 256 × 256 mm², matrix size: 256 x 256, 1 mm isotropic. All participants also underwent a diffusion-weighted MRI scan with the following parameters: TR = 7289 ms, TE = 57.1 ms, 70 slices, slice thickness: 2 mm, in-plane FOV: 220 × 220 mm², matrix size: 110 x 110, 2 mm isotropic voxels, 64 gradients at b = 1000 s/mm², and one reference b0 image. Preprocessing of the diffusion data was performed using the Elikopy pipeline[103] and included brain extraction[104], thermal denoising [Veraart et al., 2016], susceptibility-induced distortions, eddy-current distortion and head-motion correction[105] using FSL (v6.0.7.8)[106]. Due to unavailability of an unweighted diffusion image with reversed phase-encoding, Synb0-DISCO[107] was used to synthesize a distortion-free unweighted diffusion image from the T1-weighted image.

**Dual-coil paired-pulse transcranial magnetic stimulation**
General description. The dual-coil ppTMS setup involves administering a test stimulation (TS) with one coil over the primary motor cortex (M1), preceded by a conditioning stimulation (CS) delivered with another coil over a specific area of interest (i.e., TS and CS coils, respectively). This technique enables the assessment of the impact of specific cortico-cortical

and cortico-subcortico-cortical circuits on corticospinal neurons. Our objective was to investigate intra-hemispheric influences of the supplementary motor area (SMA) and orbitofrontal cortex (OFC) on M1. To achieve this, both stimulations were applied over the left, dominant hemisphere, with the TS targeted at the left M1 and the CS directed at either the left SMA or the left OFC in separate blocks of trials. We employed two small figure-of-eight coils (Magstim D25-Alpha model; wing internal diameter: 35 mm) connected to two monophasic Magstim stimulators (200² and Bistim² stimulators; Magstim, Whitland, Dyfed, UK) for delivering the TS and the CS.

TS coil location. We placed the TS coil tangentially on the scalp over the M1 area with the handle pointing backward and laterally at a 45° angle away from the midsagittal line, resulting in a postero-anterior current flow within the cortex[56,108]. To determine the optimal site for M1 stimulation, known as the "hotspot", we used markers on an electroencephalography (EEG) cap fitted on the participant's head[109–111]. Due to the close anatomical proximity of left M1 and left SMA, we had to slightly adjust the position of the M1 coil in 44 of the 47 subjects when the SMA coil was placed on the scalp; this adjusted position was used for TS during SMA blocks. Consequently, we defined two different positions for M1 stimulation: the real hotspot and an adjusted hotspot (see ref. 57, for a similar procedure).

To locate the real hotspot, we initially applied the stimulation with the center of the TS coil over the C3 location of the EEG cap (i.e., corresponding to the left M1 area)[112,113], without the presence of the CS coil on the head. We increased the stimulation intensity until consistent motor evoked potentials (MEP) were observed in the right first dorsal interosseous (FDI) muscle. We then moved the TS coil in increments of approximately 0.5 cm around this location in both the rostrocaudal and mediolateral axes. Stimulation was applied at the previously determined intensity at each new location and MEP amplitudes were visually assessed. The real hotspot was defined as the location where the largest and most consistent MEP amplitudes could be elicited. The coil was held in this position and the edges of its shape were marked on tapes placed on the EEG cap. These markings allowed us to locate the real hotspot throughout the experiment. To determine the adjusted hotspot, we positioned the CS coil over the SMA area on the head and then followed the same procedure as described above, while aiming at fitting both the TS and the CS coils on the head. After obtaining the largest MEP amplitudes, we marked the edges of the TS coil on the EEG cap. These markings enabled us to locate the adjusted hotspot during SMA blocks.

CS coil locations. To ensure precise targeting of SMA and OFC with the CS coil in each subject, we exploited neuronavigation. To do so, we used the T1-weighted high-resolution anatomical MRI data obtained in the first session (see MRI section, above). We determined the individual SMA and OFC locations on the MRI images, using Montreal Neurological Institute (MNI) coordinates in a dedicated software (Visor 2.0 Advanced NeuroTechnologies, Enschede, Netherlands). During the experiment, we relied on head and coil trackers, along with a 3D tracking device, to register the position of the CS coil with the individual MRI.

The MNI coordinates initially used to localize SMA and OFC were x = −8, y = −9, z = 77, and x = −7, y = 71, z = −4, respectively[114]. These coordinates were then slightly adjusted for each subject using the Visor software, ensuring they corresponded to the point where the scalp-to-cortex distance was minimal. Following this procedure, the final MNI coordinates for SMA and OFC locations were x = −7.9 ± 0.2, y = −5.4 ± 0.5, z = 85.6 ± 0.7, and x = −8.3 ± 0.9, y = 72.9 ± 0.4, z = 2.5 ± 2.2, respectively.

For both SMA and OFC stimulations, the center of the CS coil was placed over the corresponding target location (see Fig. 1D). In SMA blocks, the CS coil was positioned tangentially to the scalp with the handle pointing at a −100° angle away from the midsagittal line (i.e., in the counter-clockwise direction), resulting in a medio-lateral current flow within the cortex. This coil position was selected based on a previous experiment demonstrating optimal recruitment of SMA neurons. In OFC blocks, the CS coil was placed tangentially to the forefront, with the handle directed

upward and parallel to the midsagittal line, leading to a downward current flow at the cortical level.

**Stimulation intensities and inter-stimulation intervals.** Once we defined the real and adjusted hotspots, we determined the resting motor threshold (rMT) for both locations. The rMT was defined as the lowest stimulation intensity (*i.e.*, expressed as a percentage of maximal stimulator output [% MSO]) required to evoke MEPs of 50 μV amplitude on 5 out of 10 consecutive trials in the relaxed FDI muscle. The rMTs for the real and adjusted hotspots were 46.7 ± 1.1 % and 50.8 ± 1.3 % MSO, respectively. These rMT values were used to determine the stimulation intensities for the rest of the experiment. In SMA blocks, we positioned the TS coil over the adjusted hotspot and based the stimulation intensity for M1 on the rMT obtained at this location; M1 was stimulated at 120% of this rMT. Conversely, in OFC blocks, the TS coil was easily positioned over the real hotspot, allowing us to stimulate M1 at 12% of the rMT obtained for the real hotspot. Finally, the CS intensity was set at 130% and 120% of the rMT obtained over the real hotspot in SMA and OFC blocks, respectively. We decided to apply a higher stimulation intensity over SMA based on previous studies using 140% of the active motor threshold[115]. A slightly lower intensity was used over OFC to reduce facial contractions and subject discomfort.

The goal of the study was to investigate cortico-cortical and cortico-subcortico-cortical circuits between the SMA/the OFC and M1, which are assumed to be preferentially recruited with short (4–8 ms) and long (12–15 ms) inter-stimulation intervals (ISIs), respectively[57,62]. Considering that 7 ms produces the best test-retest reliability when stimulating SMA[115], we decided to use a 7 ms ISI to probe cortico-cortical circuits. To assess the influence of cortico-subcortico-cortical circuits, we decided to use an ISI of 15 ms[57]. Our main endpoint measure consisted in MEP ratios expressing CS+TS MEPs with respect to TS MEPs elicited alone.

**Blocks of trials.** The dual-coil ppTMS protocol was divided into 4 blocks of 36 trials (*i.e.*, 2 SMA blocks and 2 OFC blocks). Each block comprised trials with single-pulse (*i.e.*, TS only) and paired-pulse TMS (*i.e.*, CS+TS with the intervals mentioned above), occurring in a randomized order. Within each block, a total of 12 trials were recorded for each of the 3 conditions (*i.e.*, single-pulse, and paired-pulse with 7 and 15 ms), leading to 24 trials per condition over the whole experiment. Finally, to prevent subjects from anticipating the stimulations, we varied the intertrial interval, which ranged between 3.6 and 4.4 s (*i.e.*, rectangular distribution centered over 4 s).

### Single-coil ppTMS
**General description and coil location.** The single-coil ppTMS protocol was performed right after the dual-coil ppTMS protocol. Single-coil ppTMS involves administering a TS preceded by a CS through the same single coil, placed over M1. This technique enables the assessment of the impact of specific intra-cortical circuits on corticospinal neurons. Here, we did not use a small figure-of-eight coil because it lacks programmability and the ability to rapidly change stimulation intensity, which is necessary for single-coil ppTMS. Instead, we employed a large, programmable figure-of-eight coil (Magstim D25-Alpha model; wing internal diameter: 70 mm), which we connected to two the monophasic Magstim stimulators mentioned above ($200^2$ and Bistim$^2$ stimulators; Magstim, Whitland, Dyfed, UK) for delivering both the TS and the CS. To determine the optimal coil location in the single-coil setup, we followed the same procedure as the one utilized for identifying the real hotspot location in the dual-coil setup, employing the larger coil (*i.e.*, initial targeting of the C3 location, displacement of the coil in increments of 0.5 cm, markings on the EEG cap, *etc.*).

**Stimulation intensities and inter-stimulation intervals.** The rMT for the single-coil setup was 46.5 ± 1.1% MSO. Importantly, in single-coil ppTMS, both the intensity of the CS and the duration of the ISI play crucial roles in determining the specific subset of intra-cortical circuits under investigation. Certain variants of single-coil setups employ a subthreshold intensity for the

CS, allowing the targeting of intra-cortical circuits with low activation thresholds (Daskalakis et al., 2002). At such intensities, using a short ISI ranging from 2 to 6 ms results in the suppression of the conditioned MEP, known as short-latency intra-cortical inhibition (SICI). This phenomenon is believed to be generated by low-threshold inhibitory circuits, involving fast γ-aminobutyric acid (GABA) synapses of the GABAa subtype (Di Lazzaro et al., 2006; Ilic et al., 2002). Conversely, combining a subthreshold conditioning pulse with a longer ISI ranging from 8 to 30 ms leads to the facilitation of the conditioned MEP, referred to as long-interval intra-cortical facilitation (LICF). This effect is thought to reflect the influence of glutamatergic projections on corticospinal neurons (Ziemann et al., 1998). Lastly, when a suprathreshold conditioning pulse is combined with a long ISI ranging from 50 to 200 ms, the MEP is suppressed again, giving rise to long-latency intra-cortical inhibition (LICI). This effect is believed to be mediated by high-threshold inhibitory circuits, primarily involving slow GABAb synapses (McDonnell et al., 2006).

In this study, our objective was to investigate these three distinct types of intra-cortical circuits. To probe GABAa circuits, we utilized a subthreshold CS intensity of 80% rMT and a short ISI of 3 ms. To probe GABAb circuits, we utilized a suprathreshold CS intensity of 120% rMT and a long ISI of 100 ms. To probe glutamatergic circuits, we employed a subthreshold CS intensity of 80 % rMT and a longer ISI of 10 ms. In all three cases, we elicited MEPs using a suprathreshold TS intensity of 120% rMT, and our main endpoint measure consisted in MEP ratios expressing CS+TS MEPs with respect to TS MEPs elicited alone.

To validate that our single-coil approach effectively targeted the inhibitory and excitatory circuits under investigation, we analyzed the MEP amplitudes recorded in TS-only trials and CS+TS trials across the different stimulation intensities and inter-stimulation intervals used. In TS-only trials, the MEP amplitude was 0.56 ± 0.08 mV. In CS+TS trials, the MEP amplitudes were 0.12 ± 0.05 mV for GABAa circuits, 0.02 ± 0.03 mV for GABAb circuits, and 0.72 ± 0.10 mV for glutamatergic circuits. As anticipated, this resulted in MEP ratios below 1 for GABAa and GABAb circuits (0.36 ± 0.03 and 0.16 ± 0.03, respectively) and above 1 for glutamatergic circuits (1.26 ± 0.07). These findings confirm the expected inhibitory and excitatory circuit responses.

**Blocks of trials.** The single-coil ppTMS protocol was divided into 2 blocks of 48 trials. Here again, each block comprised trials with single-pulse (*i.e.*, TS only) and paired-pulse TMS (*i.e.*, CS+TS), occurring in a randomized order and a total of 12 trials were recorded for each of the 4 conditions (*i.e.*, single-pulse, and paired-pulse for GABAa, GABAb, and glutamatergic circuits), leading to 24 trials per condition over the whole experiment. The intertrial interval ranged between 3.6 and 4.4 s.

### Effort-based decision-making task
**Experimental set-up.** Following the TMS data acquisitions, subjects were invited to perform an effort-based decision-making task on a computer. To do so, they were seated in a comfortable chair positioned 100 cm away from the computer screen with a refresh rate of 100 Hz. Their right elbow was bent at a 90-degree angle, resting on an armrest, while they held a gripping handle with their right hand. The left hand remained free to press the left and right arrow keys on an 80 cm-high table placed in front of them. To ensure stability during contractions, a strap was used to secure the right forearm to the armrest, maintaining a consistent position.

**Task.** We employed a well-established effort-based decision-making task[6,11,13]. The task was developed using custom-made Matlab code, implemented with the Psychtoolbox library[116]. During this task, participants had to decide whether or not to exert varying levels of physical effort to earn different amounts of monetary rewards. The effort required an isometric contraction of the right biceps. Participants were instructed to attempt bringing their fist closer to their shoulder while keeping their elbow on the armrest and gripping the fixed handle. Real-time electromyography (EMG) was used to measure the strength of the contraction.

Monetary rewards were provided in euro cents, and participants were informed that all the money they earned during the task would be theirs to keep.

Each trial began with the presentation of a vertical gauge and a black horizontal line within it, indicating the amount of force the subject needed to exert. Next to the gauge, the amount of money at stake was displayed (Fig. 1B). The gauge size represented 100% of the maximal voluntary contraction (MVC), determined before the blocks of trials (see *Blocks of trials and MVC* for details). Within each trial, participants were required to exert 20, 40, 60, or 80 % of the MVC, to earn rewards of 1, 5, 10, or 20 cents, leading to 16 possible effort/reward conditions. They had 5 seconds to decide whether to engage in the contraction or not, using the right or left arrow keys on the keyboard positioned under their left hand (*i.e.*, to accept or refuse the contraction, respectively).

If they accepted the trial, a "Go" signal appeared on the screen 1 to 1.2 seconds after making their choice, indicating the start of the contraction period. Participants were provided with continuous biofeedback of their contraction level (see Fig. X), visually represented by the gauge filling. The filling of the gauge was determined by the rectified amplitude of the EMG signal recorded from the right biceps. The participants were required to maintain the contraction at least equal to the level required (*i.e.*, 20, 40, 60 or 80% of MVC) for a minimum of 2.7 seconds during a 4-second duration (*i.e.*, 2.7 seconds >66% of 4 seconds). At the end of each trial, a visuo-auditory feedback was provided to inform participants about the success or failure of their attempt and whether they earned the reward at stake. Trials where muscular contraction was detected before the "Go" signal were directly interrupted, and a "Anticipated" message appeared on the screen. The next trial began 2 seconds later, allowing participants time to recover.

If they refused the trial, the trial ended, and a blank screen appeared before the next trial started 2 seconds later. Additionally, if no decision was made during the initial 5-second period, the trial ended, and the next one began.

Blocks of trials and MVC. Subjects first performed an MVC, allowing us to calibrate the force level required during the effort-based decision-making task. During the MVC assessment, we verbally encouraged them to flex their arm as strong as possible during 5s for three consecutive repetitions with a 30s rest in-between. The MVC was considered as being the maximal contraction achieved during one of the three repetitions, with the mean between the second and fourth second being taken as the result for each repetition.

Participants then performed a practice block of 16 trials, so that they could get used to the task and have an idea of the force required to reach the different percentages of their MVC. Once the familiarization completed, they performed 5 blocks of 32 trials, each block including two repetitions of each of the 16 effort/reward conditions. There were therefore 160 trials in total, with 10 trials per condition. Conditions were presented in a randomized manner within each block.

A new MVC was measured 30 s after the end of each block. A rest period of 90 s was given to the subject following the MVC acquisition, after which the next block was realized. Hence, the MVC was assessed 6 times during the experiment: before each of the 5 blocks and after the last one. We exploited the MVC so that the required force during each block was readjusted with regards to the putative fatigue in each subject. Importantly, subjects were not aware that the required force during the blocks was readjusted based on the last MVC recorded.

Questionnaires. In the final part of session 2, participants completed three questionnaires on a computer, implemented using the Psytoolkit software. These questionnaires aimed to measure inter-individual differences in three key variables of interest for our study: apathy, depression and anhedonia, as detailed below.

Apathy. We utilized the extended version of the Lille Apathy Rating Scale (LARS-e), adapted for a healthy population by Bonnelle et al. (2015). The LARS-e consists of 51 items, with 5 propositions for each item, and grades ranging from 1 to 5 for each item, reflecting the subject's level of motivation (a higher number indicating higher motivation). The apathy score is calculated by subtracting the score of each item from the maximum score of 5, then summing up the corresponding items. The obtained sum is divided by the number of items taken, resulting in a mean score between 1 and 5, with a higher score indicating a higher level of apathy[17] (see *Data processing* section below). While recent work has proposed the construction of composite trait apathy scores using factor analysis across multiple questionnaires[117], a large and robust body of literature currently uses single-scale scores alone as a validated and sensitive measure of apathy-related motivational states, particularly when combined with task-based behavioral assessments to isolate specific motivational components. Our study followed this well-established approach to directly relate inter-individual variability in apathy to structural and effective connectivity, while acknowledging that no single method can capture the full complexity of motivational processes.

Depression. Given that apathy is highly prevalent in depression[13] and that depression is associated with changes in MEP amplitudes[118], it was crucial to assess the level of depression to control for potential confounding factors when examining the relationship between MEP amplitudes and apathy scores. For this purpose, we utilized the Depression Anxiety Stress Scales (DASS 21; as in ref. [17]), a self-report questionnaire comprising 3 subscales: depression, anxiety, and stress. Each subscale contains 7 items, and each item has 4 propositions, with scores ranging from 0 to 3. The total score for each subscale is obtained by summing up the corresponding items.

Anhedonia. Anhedonia is defined as the loss of the ability to experience interest or pleasure in activities that were previously enjoyed. While apathy and anhedonia are two distinct but closely related syndromes, both associated with amotivation, some studies have shown a positive correlation between these two syndromes in certain conditions such as Parkinson's disease and schizophrenia[13]. Thus, it was essential to ensure that any relationship between MEP amplitudes and apathy scores was specific to apathy and did not concern anhedonia. To measure anhedonia, we used the French version of the Snaith-Hamilton Pleasure Scale (SHAPS)[119]. The SHAPS consists of 14 items, and each item is assigned either 0 or 1 point. The total score ranges from 0 to 14, reflecting the level of anhedonia.

### Data processing
Questionnaires data processing. The responses of all participants to each item of the questionnaire were recorded and stored in the Psytoolkit software, and later exported to R (R Core Team, 2020) for further analysis. We calculated the total score for both the LARS-e and the SHAPS. In addition, we derived behavioral, emotional, and cognitive apathy subscores from the LARS-e: Action Initiation (behavioral apathy), Emotional Response (emotional apathy), and Intellectual Curiosity–Novelty (used as a proxy for cognitive apathy). These subscore were exploited in a second-level analysis to test whether some of the circuits identified in our first-level (i.e., LASSO) analysis were associated with specific dimensions of apathy. Regarding the DASS-21, we computed only the score for each subscale (depression, anxiety, and stress) since the objective of this scale is not to provide a summation of all the items. We focused on the depression score in the statistical analyses. The distribution of apathy scores is provided in Fig. 1A, while distributions of anhedonia and depression scores are provided in Supplementary Fig. 1.

Computation phenotyping of decision-making. We performed computational phenotyping of individuals' behavior in an effort-based decision-making task to estimate inter-individual differences in effort and reward valuation. Our model yielded us an estimation of how these variables influenced the value of an offer for each participant. Based on previous literature[8,10,120], we tested candidate models of value computation, using standard methods (minimization of Bayesian information criterion (BIC) and visual inspection of model fits, see Supplementary Fig. 3). More specifically, for each trial, the model computes a subjective

value of the offer that depends on the proposed effort and reward. This value is then transformed to an estimated probability of acceptance (between 0 and 1) based on a softmax function[8]. Parameter estimates were determined with the function fmincon on Matlab 2022b (Matlab, Mathworks, USA) with the goal to minimize the difference between each participant's modelled and real choice data (*i.e.*, to minimize the negative log-likelihood of the model given the data). The models differed in the form of the cost function (linear, quadratic, or exponential effort discounting) and in the inclusion or absence of a $\beta_{Time}$ parameter capturing trial-wise adaptation (*i.e.*, modulation of value computation by trial number). The winning model was the following:

$$Value = \beta_{Effort} \times EFFORT_{LEVEL}.^2 \times (1 + \beta_{Time} \times t) + \beta_{Reward} \times REWARD_{LEVEL} + \beta_0$$

Here, the value of an offer depends on a quadratic function of the proposed $EFFORT_{LEVEL}$ and on the available $REWARD_{LEVEL}$ (modulated by the $\beta_{Effort}$ and $\beta_{Reward}$ parameters, respectively). To account for potential time-on-task effects on value computation (due to the repetition of trials), this model also includes a $\beta_{Time}$ parameter that linearly modulated the cost term based on the trial number[10]. $\beta_0$ represents the baseline tendency to accept an offer (the value of accepting an offer for 0 reward). Following model fitting, we obtained for each subject $\beta_{Effort}$ and $\beta_{Reward}$ parameters, which we took as proxy of effort and reward valuation, respectively. As such, in the above equation, the more negative the $\beta_{Effort}$ is, the stronger effort valuation is, indicating a steeper relative decline in acceptance rates with increasing effort. Of note, due to the quadratic nature of the cost function in the winning model (*i.e.*, $\beta_{Effort} \times EFFORT_{LEVEL}^2$), this effect is particularly pronounced at high effort levels: the more a participant's acceptance rate drops for high efforts (*e.g.*, 80% MVC), the steeper the curvature of the effort-valuation function and the more negative the $\beta_{Effort}$. Conversely, the more positive $\beta_{Reward}$ was, the stronger reward valuation was (*i.e.*, the stronger the increase of acceptance rates as a function increasing reward levels). Also, to facilitate interpretation of the correlations, $\beta_{Effort}$ values were multiplied by −1 before proceeding with statistical analyses, such that positive $\beta_{Effort}$ values reflected stronger effort valuation.

**Streamline tractography analysis**. We utilized the multi-shell, multi-tissue constrained spherical deconvolution (MSMT-CSD) algorithm[121] for local modeling. Streamlines were obtained using the iFOD2 tractography algorithm from MRTrix3 (v3.0.3)[122] with the following parameters: 100,000 streamlines, a maximum angle of 15°, a step size of 1 mm, and a cutoff value of 0.1. The tractograms were then filtered down to 50,000 streamlines using the SIFT algorithm to improve the fit between the density of the streamlines generated and the diffusion signal[54].

For structural connectivity analysis, we used the human Brainnetome atlas [Fan et al., 2016]. The registration of the atlas was conducted using a two-step process: first, a rigid registration aligned the diffusion-weighted MRI data to the T1-weighted MRI subject space. Next, a non-rigid transformation was applied to align the T1-weighted subject space with the MNI template space. This procedure utilized a Python implementation of the ANTS algorithm [Avants et al., 2008]. The transformations obtained were used to register the atlas ROIs to the subject diffusion MRI space, followed by a 2 mm dilation of the ROIs to include the white-gray matter interface. Connectivity matrices were constructed from the filtered tractograms, representing all connections between each pair of structures in the atlas. Brain structures were considered connected if any part of a streamline passed through both ROIs. Given that the number of streamlines is an estimation that can fluctuate from one iteration to another, we run the full analysis for 50 iterations, resulting in 50 connectivity matrices per subject. We then calculated the average number of streamlines for each pair of structures out of the 50 iterations to obtain an average connectivity matrix for each subject. This approach allowed us to obtain a number of streamlines that was not specific to a given iteration. A higher number of streamlines

between two structures in the connectivity matrix (*i.e.*, after correction by SIFT) in one subject compared to another, reflects a greater probability that the fibers linking these two regions are more numerous, occupy a larger volume, or are in better condition, thus indicating greater structural connectivity.

As for our TMS data, we focused on the left hemisphere and selected pairs of structures from the matrices that comprised the fronto-BG-M1 and fronto-M1 circuits of interest in the present study. Specifically, we examined the following structures: the SMA, the OFC, the M1, and all structures of the BG that were identifiable using the Brainnetome atlas. These included the dorsal Caudate (dCaudate), the ventral Caudate (vCaudate), the dorsolateral Putamen (dlPutamen), the ventromedial Putamen (vmPutamen), and the Nucleus Accumbens (NAcc). To ensure the completeness of the circuits, we also incorporated the motor part of the thalamus. We extracted the number of streamlines for the following 19 pairs of structures: SMA-M1, SMA-dCaudate, SMA-vCaudate, SMA-dlPutamen, SMA-vmPutamen, SMA-NAcc, OFC-M1, OFC-dCaudate, OFC-vCaudate, OFC-dlPutamen, OFC-vmPutamen, OFC-NAcc, dCaudate-GP, vCaudate-GP, dlPutamen-GP, vmPutamen-GP, NAcc-GP, GP-Thalamus, and Thalamus-M1. Our objective was to determine whether any of these pairs exhibited an association in terms of structural connectivity (*i.e.*, of number of streamlines) with apathy scores, $\beta_{Effort}$, and $\beta_{Reward}$.

**TMS data processing**. The EMG data were analyzed using custom Signal and R scripts (R Core Team, 2020). We applied the same data processing procedure on MEPs obtained in dual-coil and in single-coil ppTMS blocks. First, trials in which the root mean square of the EMG signal exceeded 200 µV before stimulation (*i.e.*, −250 to −50 ms from the pulse) were excluded from the analysis. Second, MEPs with an amplitude exceeding 3 SD around the mean within a given condition were also discarded to reduce MEP variability effects. After applying this cleaning procedure, we retained an average of 22 trials for each condition. The minimum and maximum numbers of MEPs per condition was 17 and 24, respectively. Third, we computed the mean peak-to-peak MEP amplitude for each subject and condition. Finally, we computed a ratio expressing MEP amplitudes elicited by the TS in paired-pulse trials relative to MEP amplitudes elicited by the TS in single-pulse trials. This procedure yielded one MEP ratio for each subject and condition (i.e., for short and long-latency SMA and OFC circuits in dual-coil blocks and for GABAa, GABAb and glutamatergic circuits in single-coil blocks). Ratios above 1 were considered indicative of a facilitatory influence of the CS on MEP amplitude, while ratios below 1 were interpreted as reflecting a suppressive effect. Our objective was to determine whether any of these measures of effective connectivity exhibited an association with apathy scores, $\beta_{Effort}$, and $\beta_{Reward}$.

**Statistics and reproducibility**

All statistical analyses were performed using custom-made R scripts (R Core Team, 2020) on our sample of 45 subjects. Our goal was to determine which independent variables quantifying structural connectivity (*i.e.*, the average numbers of streamlines in the tracts of interest) and effective connectivity (*i.e.*, MEP ratios) would be associated with our three dependent variables of interest: apathy scores, $\beta_{Effort}$ and $\beta_{Reward}$. To address this issue, we exploited a conservative two-step statistical approach.

**Step 1: independent variable selection with LASSO regression**. First, we utilized Least Absolute Shrinkage and Selection Operator (LASSO) regression to perform independent variable selection. LASSO shrinks some regression coefficients to zero, thus eliminating less relevant independent variables.

The LASSO regression can be described by the following equation:

$$\hat{\beta} = \underset{\beta}{\operatorname{argmin}} \left\{ \frac{1}{2n} \sum_{i=1}^{n} (y_i - x_i^T \beta)^2 + \lambda \sum_{j=1}^{p} |\beta_j| \right\}$$

Where $y_i$ represents the response variable for the $i$-th observation, $x_i$ is the vector of independent variables for the $i$-th observation, $\beta$ is the vector of regression coefficients, $n$ is the number of observations, $p$ is the number of predictors, $\lambda$ is the regularization parameter controlling the shrinkage of the regression coefficients associated with each independent variable.

The first term, $\frac{1}{2n}\sum_{i=1}^{n}(y_i - x_i^T\beta)^2$, represents the residual sum of squares, which measures the fit of the model (*i.e.*, as in classical regression model, without regularization term). The second term, $\lambda\sum_{j=1}^{p}\left|\beta_j\right|$ is the regularization term, which penalizes the absolute values of the coefficients $\beta_j$, shrinking less important regression coefficients towards zero as $\lambda$ increases.

To estimate the optimal value of the regularization parameter $\lambda$, a 10-fold cross-validation is classically exploited in the following manner: 1. Partitioning the data: the dataset was randomly split into 10 equal-sized subsets (folds). 2. Training and validation: for each fold, the model was trained on 9 folds (training set) and tested on the remaining fold (validation set). This process was repeated 10 times, each time using a different fold for validation. 3. Model evaluation: for each iteration, the mean squared error (MSE) was calculated to measure the model's predictive accuracy. 4. Averaging the errors: the MSE was averaged across all folds to compute the overall cross-validated error for each $\lambda$ value. 5. Optimal $\lambda$ selection: the $\lambda$ value that minimized the average cross-validated MSE was selected as the optimal parameter. Finally, given that the $\lambda$ parameter is an estimation that can fluctuate from one iteration to another, we run the cross-validation for each of the six LASSO regressions for 100 iterations and calculated the average $\lambda$ parameter out of the 100 iterations.

Once the average optimal $\lambda$ value was determined (the one that minimizes MSE), the LASSO regression is classically refit on the dataset using this $\lambda$ value. During this final fitting step, the LASSO algorithm thus selects the $\beta$ coefficients that balance the trade-off between minimizing MSE and penalizing larger $\beta$ coefficients. Some $\beta$ values will be shrunk to zero, meaning the corresponding independent variables are excluded from the regression model. Non-zero $\beta$ values correspond to the most important independent variables that contribute to explaining the dependent variable (*i.e.*, apathy scores, $\beta_{\text{Effort}}$, and $\beta_{\text{Reward}}$ in the present study).

**Step 2: Partial correlations controlling for potentially confounding variables.** Once relevant independent variables selected by the LASSO regression, we performed partial correlations between the dependent variables and the selected independent variables. This was done to control for multiple potentially confounding variables that may covary with apathy, ensuring that the associations we draw between structural or effective connectivity and apathy are specific to apathy. The control variables in our study included age, gender, depression score, and anhedonia score. The partial correlation provided an R-value and a $p$-value for each correlation. We set the significance level at $p < 0.05$.

**Step 3: Replicability of the main findings.** The absence of association between the independent and dependent variables identified by the LASSO regression was further confirmed through Bayesian analyses. Additionally, the exact same variable selection as the one operated by the LASSO regression was obtained using a less conservative Elastic Net regression, replicating exactly the same results using a different statistical approach and confirming the absence of false negatives (Supplementary Fig. 4). Finally, all variables selected by the LASSO regression were also chosen using a stepwise regression approach, again confirming that the findings are independent of the analytical method used.

**Reporting summary**
Further information on research design is available in the Nature Portfolio Reporting Summary linked to this article.

**Data availability**
Datasets are freely available on the Open Science Framework repository upon publication, at: https://osf.io/a59mt/files/osfstorage. Source data underlying the main graphs of each figure are provided as Supplementary Data files (Supplementary Data 1-6). Any additional information required to reanalyze the data reported in this work paper is available from the lead contact upon request.

**Code availability**
Custom codes generated during this study are freely available on the Open Science Framework repository upon publication, at: https://osf.io/a59mt/files/osfstorage. Any additional information required to reanalyze the data reported in this work paper is available from the lead contact upon request.

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

## Acknowledgements
G.D. was supported by a FNRS Research associate grant. P.V. was supported by the Fund for Research training in Industry and Agriculture (FRIA/FNRS; FC29690) and grants by the Platform for Education and Talent (Gustave Boël—Sofina Fellowships), Wallonie-Bruxelles International. J.D. was supported by three FNRS research grants (T008219F, J005921F and T007023F). We would like to thank Matthieu Boisgontier, Ahmad Nour-eddine and Valentin Touzé for their participation in data acquisition.

## Author contributions
G.D.: study conception, data acquisition, T.M.S. data processing, statistical analysis, interpretation of the results, article writing from the first draft to the final version. PV: computational modelling, interpretation of the results, article writing. L.D., Q.D., N.D.: M.R.I. data processing, article writing. A.Z.: computational modelling, interpretation of the results, article writing. J.D.: study conception, interpretation of the results, article writing.

## Competing interests
The authors declare no competing interests.
