## [Transparent Peer Review file · Communications Biology]

Fronto-motor circuits linked to effort-based decision-making and apathy in healthy subjects

Corresponding Author: Dr Gerard Derosiere

Version 0:

Reviewer comments:

Reviewer #1

(Remarks to the Author)

This is a very thorough investigation of structural and effective connectivity in fronto-M1, fronto-BG-M1, and intra-M1 circuits in relation to apathy and effort-based decision-making. Using computational modeling of effort and reward valuation during a decision-making task, the authors show that apathy scores are associated with both structural and effective connectivity in fronto-M1 and fronto-BG-M1 circuits. Circuits originating from the supplementary motor area were primarily related to effort valuation, while intra-M1 GABAergic connectivity correlated exclusively with reward valuation.

This is a very carefully conducted experiment on a pathology whose underlying neural bases are still largely misunderstood. The approach is very ambitious, relating apathy scores in a healthy cohort to structural and functional connectivity with frontal, BG and M1 circuits. Across the board, the methods are carried out according to the highest standards (decision making test, questionnaires, DWI, TMS) and are analyzed with the proper statistical tools (computational phenotyping, LASSO, elastic net validation). Enough details are provided for the work to be replicated. The results are interesting and the interpretations are generally sound. The paper is well written, with a high level of structure which allows to convey rather clearly the numerous relationships that are investigated. Overall this paper is certainly one of the most complete assessments of M1 structural and functional connectivity in the context of decision making, and should spark interesting lines of inquiry into the circuit-level dysfunctions that might mediate apathy.

The introduction might benefit from a more fleshed out review of what is known regarding the neural bases of apathy. On page 4, the authors really only say that apathy is tied to “increases and decreases in volume and activity” and “increases or decreases in white matter connectivity” of fronto-BG regions. This leads to a very exploratory paper with no hypothesized direction of the expected effects. I appreciate that the authors do discuss their findings in light of previous work in the discussion (top of p24), but any intuition that could be built may help the reader make more sense of the results as they come along.

The downside of exploring so many variables at once is that it is more difficult to synthesize the main claim of the paper. The last sentence of the abstract and the intro, for instance, are quite general and leave it to the reader to extract what the primary contribution of the paper is. I do commend the authors for Fig7, but still if they could distill the main (most expected, most robust) findings and emphasize them in text form, it certainly would help.

One limitation of the methods used is that little is known regarding the normative relationships that should exist between the structural and functional connectivity measures, and how they relate to behavioural phenomena. For instance, it is not obvious whether strong connectivity is a good or bad thing. A symptom of this is page 24, where high connectivity is proposed to be either detrimental or helpful. I guess at this point it is a valid contribution to simply characterize these relationships, and leave it to future studies to test more directional hypotheses, but still if the authors could find a way to build intuition the paper would read less like just a collection of results whose interpretation could go either way.

I may have missed it but do the authors explain the anticipated direction of the short- and long-latency OFC-M1 ppTMS?

Does OFC have a facilitatory or inhibitory influence on M1?

Figure 2 caption p11: “The green bars represent...”

P39: “We performed computational phenotyping...”

P40: “the stronger effort valuation IS...”

Fig S2 caption: “The correlogram (LEFT panel)...”

Reviewer #2

(Remarks to the Author)

In this manuscript, Derosiere and colleagues investigated potential structural and functional changes underlying apathy, focusing on the frontal cortex, primary motor cortex (M1), and basal ganglia. They used MRI-based tractography and transcranial magnetic stimulation (TMS) to examine these regions. The study involved a large group of healthy subjects performing an effort-based decision-making task. The authors correlated structural and effective connectivity data with computational parameters derived from the task and with apathy scores. They found clear correlations between apathy scores and specific circuit connectivity and provided evidence for differential contributions of these circuits to effort versus reward valuation.

Overall, I found the manuscript to be very well written and easy to follow. To the best of my knowledge, the results are novel and appear robust due to careful data analysis. I have only minor comments. I would like to disclose that while I am very interested in the mechanisms underlying decision-making and motor control in the context of effort-based tasks, I am not a specialist in human behavior or the imaging techniques used in this study.

Comments:

#In the introduction, the authors refer to the basal ganglia (BG) in general. However, different regions of the BG have not been equally implicated in effort/reward decision-making. It would be helpful if the authors could discuss this more specifically in the introduction.

#For a naive reader, it would be nice to at least provide a couple of sentences regarding intra-M1 GABA_A, GABA_B, and glutamatergic circuits connectivity can be inferred from such imaging techniques

#In the results section, there was little practical information about the decision-making task, yet the authors derived effort and reward sensitivities from it and examined their correlation with imaging data. For a naive reader, it would be helpful to provide a clearer description of the task structure in the main text, rather than requiring reading the methods. More generally, I wondered whether it was appropriate to limit the description of this key component to a schematic overview in Figure 1 and Supplementary Figure 2. A more detailed presentation of the behavioral results in the main figures or text might improve comprehension of this study.

#Related to the point above, is there no possibility to use movement related data to infer apathy? One could expect that subjects with a higher apathetic score would exhibit "difficulty" in generating the strongest force for the lowest reward. Or does the visual feedback on strength level prevent this kind of effect?

#One intriguing result is that the computational variables derived from the decision making task do not correlate with apathy scores. Could it be that the task used in the lab does not require sufficient effort or involve sufficiently vital rewards? This could be discussed further.

Typos:

In the method section: As such, in the above equation, the more negative the β Effort is, the stronger effort valuation *iss*
Supplementary Figure 1: the last sentence of the legend is incomplete.

Signed David Robbe

Reviewer #3

(Remarks to the Author)

This paper explores the neural correlates of apathy and effort-based decision-making in healthy individuals, focusing on connectivity within fronto-basal ganglia (BG)-motor circuits. Using a multimodal approach combining diffusion-weighted MRI tractography and paired-pulse transcranial magnetic stimulation, the researchers assessed both structural and effective connectivity among 45 participants. Behavioral data were collected via an effort-based decision-making task, analyzed with computational modeling to extract sensitivity to effort (β Effort) and reward (β Reward). Apathy levels were assessed with validated questionnaires. They found that apathy scores positively correlate with both structural connectivity (in SMA-M1, SMA-putamen, and OFC-caudate tracts) and effective connectivity (in long-latency OFC-M1 circuits). Effort sensitivity was linked to structural connectivity in SMA-M1 and SMA-NAcc pathways but not to effective connectivity. Conversely, reward sensitivity was associated with structural connectivity in the NAcc-GP tract and effective connectivity in intra-M1 GABAergic circuits. Interestingly, the study found no correlation between apathy or valuation measures and M1 net output, suggesting that connectivity—rather than output strength—is the key driver. The manuscript is well written and methodologically thorough, with findings highlighting the distinct fronto-motor circuits underlying apathy, effort, and reward valuation, and potential implications for targeted neuromodulation strategies in clinical populations. However, before considering this work for publication, there are some theoretical and methodological issues that should be addressed. My main concerns include interpretations of the results of the one side and the model fitting on the other, plus some minimum polishing/minor points.

1. Authors present an interesting study of the apathy phenomenon as a network-level-phenomenon involving fronto-basal ganglia circuits and their connectivity to motor regions. However, several clarifications and enhancements could strengthen the framing of the study, still partially lacking clarity and motivation. First, the construct of apathy is introduced as a unitary concept, whereas the literature consistently distinguishes between behavioral, emotional, and cognitive subtypes (for instance Levy & Dubois, 2006 <https://doi.org/10.1093/cercor/bhj043>; Radakovic & Abrahams, 2014; <https://doi.org/10.1016/j.psychres.2014.06.010>). Given the use of an effort-based decision-making task, it would be helpful for the authors to explicitly situate their work within the framework of behavioral or motivational apathy and reference prior work that supports this dimensional view. Second, while the authors argue that apathy varies in healthy individuals, they could more clearly articulate the relevance of studying subclinical apathy by citing empirical work on its neural correlates in

non-clinical populations (like Bonnelle et al., 2015 <https://doi.org/10.1093/brain/awv278>; Grogan et al., 2018, <https://doi.org/10.1016/j.neuropsychologia.2018.05.008>). Additionally, the rationale for including M1 in the apathy network is compelling but underdeveloped; the introduction would benefit from theoretical grounding—such as distributed decision-making or embodied cognition models—to explain how M1 might contribute to motivational processes beyond simple movement execution (Cisek & Kalaska, 2010, <https://doi.org/10.1146/annurev.neuro.051508.135409>). Finally, the authors may consider refining the final paragraph of the introduction to better emphasize the specific research questions, reduce detailed methodological content, and more clearly highlight the novel contribution of their multimodal approach. Together, these revisions would sharpen the conceptual framing and improve clarity for a broader audience.

2. While the study compellingly demonstrates behavioral correlations between apathy scores and task performance in high-effort/high-reward trials, the lack of significant associations between apathy and the model-derived β Effort and β Reward parameters raises important concerns about the sensitivity or specificity of the current computational model. This disconnect is particularly noteworthy given the central goal of the study—to characterize apathy in terms of latent motivational constructs. The reliance on static, two-parameter models (e.g., β Effort and β Reward) may oversimplify the decision-making process, especially when applied to a heterogeneous construct like apathy that likely involves fluctuating cognitive and affective states (e.g., fatigue, learning, internal variability in valuation). As such, while the current modeling approach follows precedent, it might not capture the full complexity of motivational dynamics in real-time decision-making. Authors should consider incorporating additional parameters into the computational model that reflect decision noise or exploration/exploitation tradeoffs (e.g., softmax temperature or lapse rate); effort discounting curves (nonlinear or parabolic rather than linear); fatigue accumulation or trial-wise adaptation (dynamic models with trial history). Also, they could provide a post-hoc mediation by examine whether task behavior mediates the relationship between apathy scores and connectivity, which might clarify whether the lack of correlation with β parameters is a modeling artifact or a real dissociation.

3. When authors consider the past two decades of “progress in neuroimaging and brain stimulation have enabled new ways to explore this idea in humans. Advanced analyses of diffusion weighted imaging (DWI) such as streamline tractography allow quantifying how structural connectivity in specific circuits varies among individuals^{48,49}. Additionally, paired-pulse transcranial magnetic stimulation (ppTMS) allows pre-activating certain circuits and measuring the impact of this pre-activation on neural activity in M1's corticospinal neurons^{38,50,51}, providing a measure of how effective connectivity in circuits projecting to these neurons varies among individuals. Integrating these complementary methods through a multimodal framework thus provides a unique opportunity to investigate the different circuits connecting SMA, OFC and the BG to M1 in a comprehensive way” they missed some recent literature that already started exploring the DWI-TMS combination in the context of action decision and reviewing network including SMA and other frontal and premotor areas (Tagliaferri et al., 2023 <https://doi.org/10.1523/JNEUROSCI.0406-23.2023>, Tagliaferri et al., 2024 <https://doi.org/10.1007/s00429-024-02778-4>)

Finally, author measure apathy via self-report (LARS-e), which may not fully capture behavioral or neurobiological manifestations. A growing consensus suggests combining self-report tools like LARS-e with behavioral paradigms (e.g., EEfRT, grip-force tasks - 10.1001/jamapsychiatry.2017.2901; 10.1038/s41583-018-0029-0) and EMA (10.1037/abn0000191; 10.1016/j.schres.2021.02.005) to provide a more accurate, multidimensional measure of apathy. This triangulation can help disentangle motivational deficits from reporting biases or mood symptoms. I suggest including some objective behavioral tasks or ecological momentary assessments to complement the questionnaire, and as a minimum to incorporate the limitations of this approach.

Minor suggestion/improvement: as I suggested in point 1, while M1 involvement is novel and well-justified, in the end the authors might have overemphasized its role without considering broader network contributions in the explanation of the results beyond the tractography. They could take into deeper consideration functional connectivity or broader fMRI networks (e.g., default mode or salience networks) that may also contribute to apathy.

Version 1:

Reviewer comments:

Reviewer #1

(Remarks to the Author)

The authors have adequately revised the manuscript. It is now suitable for publication. Good job.

Just 1 typo in the Sup Fig5 caption 3rd line. It should read "The leftmost column"

Reviewer #2

(Remarks to the Author)

I thank the authors for having carefully addressed all my comments.

Reviewer #3

(Remarks to the Author)

The authors addressed all my comments and concerns, and after all the requested reviews, the paper has significantly improved. I have no other comments.

Revision of Manuscript #COMMsBIO-25-3155 entitled “Fronto-motor circuits linked to effort-based decision-making and apathy in healthy subjects” by Gerard Derosiere, Pierre Vassiliadis, Laurence Dricot, Quentin Dessain, Nicolas Delinte, Alexandre Zénon, and Julie Duque.

We sincerely thank the three reviewers for their thoughtful and encouraging evaluations of our work. We are pleased that they found the study to be novel, rigorous, and relevant. We have carefully addressed all comments and incorporated the suggested revisions, which we believe have further strengthened the manuscript. We trust that the revised version will meet the expectations of both the reviewers and the editors. All changes are highlighted in blue in the revised manuscript. Please find our detailed point-by-point replies below.

Reviewers' comments:

Reviewer #1 (Remarks to the Author):

This is a very thorough investigation of structural and effective connectivity in fronto-M1, fronto-BG-M1, and intra-M1 circuits in relation to apathy and effort-based decision-making. Using computational modeling of effort and reward valuation during a decision-making task, the authors show that apathy scores are associated with both structural and effective connectivity in fronto-M1 and fronto-BG-M1 circuits. Circuits originating from the supplementary motor area were primarily related to effort valuation, while intra-M1 GABAergic connectivity correlated exclusively with reward valuation.

This is a very carefully conducted experiment on a pathology whose underlying neural bases are still largely misunderstood. The approach is very ambitious, relating apathy scores in a healthy cohort to structural and functional connectivity with frontal, BG and M1 circuits. Across the board, the methods are carried out according to the highest standards (decision making test, questionnaires, DWI, TMS) and are analyzed with the proper statistical tools (computational phenotyping, LASSO, elastic net validation). Enough details are provided for the work to be replicated. The results are interesting and the interpretations are generally sound. The paper is well written, with a high level of structure which allows to convey rather clearly the numerous relationships that are investigated. Overall this paper is certainly one of the most complete assessments of M1 structural and functional connectivity in the context of decision making, and should spark interesting lines of inquiry into the circuit-level dysfunctions that might mediate apathy.

Authors' answer: Thank you very much for your enthusiastic feedback on our article and the study's methodology and findings.

The introduction might benefit from a more fleshed out review of what is known regarding the neural bases of apathy. On page 4, the authors really only say that

apathy is tied to “increases and decreases in volume and activity” and “increases or decreases in white matter connectivity” of fronto-BG regions. This leads to a very exploratory paper with no hypothesized direction of the expected effects. I appreciate that the authors do discuss their findings in light of previous work in the discussion (top of p24), but any intuition that could be built may help the reader make more sense of the results as they come along.

Authors’ answer: We thank the reviewer for this comment. We have expanded the Introduction to better synthesize existing evidence on the neural bases of apathy. We clarify that grey matter findings most frequently report decreases in volume or activity within fronto-BG structures, though increases are also occasionally observed. We also specify the roles of individual regions (e.g., SMA, putamen, OFC, nucleus accumbens, globus pallidus). For white matter, we now explicitly note that the literature is mixed in terms of directionality (increases or decreases in connectivity), likely due to differences in apathy severity, in the underlying pathology, or in the circuits examined. We now refer to the review of Le Heron et al. (*Neuropsychologia*, 2018) highlighting these mixed patterns. Given these inconsistent findings and the low amount of data on motor circuits, we had indeed no a priori hypothesis regarding directionality of the relationship with apathy and effort and reward valuation. Hence, we do not feel comfortable with presenting a hypothesis formulated a posteriori and decided not to do so. Rather, we have now made explicit that our study hypothesized altered connectivity within the studied circuits, without making directional assumptions regarding connectivity–behavior relationships.

Specifically, the Introduction now reads (lines 68-85): “...*Whether clinical or subclinical*^{17,18}, *apathy is tied to alterations consisting predominantly of decreases*¹³, *though occasionally of increases, in the volume and activity of distributed fronto-basal ganglia (BG) structures known for their role in deciding to exert efforts for rewards*^{13,24,25}. *These structures include the supplementary motor area (SMA)*¹⁸ *and the putamen*²⁹, *involved in effort valuation*³⁰⁻³², *the orbitofrontal cortex (OFC)*³³, *engaged in reward valuation*^{34,35}, *and the nucleus accumbens*^{36,37}, *thought to weigh up efforts and rewards*^{24,38}. *Alterations are likewise observed in downstream BG nuclei, such as the globus pallidus*²², *where effort and reward computations have also been reported*^{39,40}. *Beyond these grey matter alterations, recent studies show that apathy is associated with altered white matter connectivity between fronto-BG structures*^{18,26-28}, *though the direction of these changes – whether increases or decreases – varies across studies, likely reflecting heterogeneity in apathy severity, study populations, or the specific circuits examined (see*⁴¹, *for review). This pattern suggests that, relative to individuals without apathy, both hypo- and hyper-connectivity in different components of the fronto-BG circuitry can be associated with apathy. ...”*

AND, in the last paragraph of the Introduction we added (lines 107-133): “*In this study, we aimed to characterize how structural and effective connectivity within fronto-striato-motor circuits relates to individual differences in apathy in the general population. Specifically, we asked whether distinct circuits are differentially associated with apathy levels, as measured through questionnaires*^{18,19}, *and the valuation of efforts and rewards, as measured through computational phenotyping of behavior in an effort-based decision-making task*^{10,11,60,61}. *While previous studies have linked apathy to*

altered connectivity within fronto-BG networks, findings have been mixed in terms of directionality – reporting both increases and decreases in connectivity – and largely focused on non-motor circuits. In light of these mixed findings and the paucity of data on motor circuits¹⁸, the present study hypothesized altered connectivity within these circuits, without making directional assumptions regarding connectivity–behavior relationships. ...”.

The downside of exploring so many variables at once is that it is more difficult to synthesize the main claim of the paper. The last sentence of the abstract and the intro, for instance, are quite general and leave it to the reader to extract what the primary contribution of the paper is. I do commend the authors for Fig7, but still if they could distill the main (most expected, most robust) findings and emphasize them in text form, it certainly would help.

Authors’ answer: We fully agree with the reviewer. To address this point, we have revised the final paragraph of the Introduction to explicitly distil the strongest findings and their relevance (lines 125-133): *“Among the most robust findings, we observed that both OFC-BG structural connectivity and OFC-M1 long-latency effective connectivity were positively associated with apathy scores, thus replicating this relationship across our two connectivity measures. Moreover, SMA-M1 structural connectivity was related to both apathy and effort valuation, consistent with SMA’s known role in effort processing^{30,31}. Altogether, these findings suggest that distinct fronto-motor circuits – originating from both OFC and SMA – contribute to apathy and effort valuation, highlighting complementary circuits as potential neuromodulatory targets for specific mechanisms of disrupted motivated behavior.”.* We underscore them again at multiple locations in the Results section to guide the reader toward the main conclusions and we have similarly revised the legend of Figure 7 to reinforce these key findings (lines 582-588): *“Figure 7: Summary of the main correlations uncovered in the present study. A. Detailed correlations. [...]. Notably, both OFC–BG structural connectivity and OFC–M1 long-latency effective connectivity showed positive correlations with apathy scores (and, more specifically, behavioral apathy; see Supplementary Figure 5). Hence, this relationship was replicated across two independent modalities (DWI and dual-coil ppTMS). In addition, SMA–M1 structural connectivity was associated with both apathy scores and effort valuation, which is highly consistent with the SMA’s established role in effort processing. B. [...].”*

One limitation of the methods used is that little is known regarding the normative relationships that should exist between the structural and functional connectivity measures, and how they relate to behavioural phenomena. For instance, it is not obvious whether strong connectivity is a good or bad thing. A symptom of this is page 24, where high connectivity is proposed to be either detrimental or helpful. I guess at this point it is a valid contribution to simply characterize these relationships, and leave it to future studies to test more directional hypotheses, but still if the authors could find a way to build intuition the paper would read less like just a collection of results whose interpretation could go either way.

Authors' answer: We agree with the reviewer that, at present, the field lacks a normative framework for interpreting whether higher connectivity should be considered adaptive or maladaptive in the context of apathy. This uncertainty is reflected in the heterogeneity of prior findings, with both increases and decreases in connectivity associated with apathy across different populations and circuits. We have clarified this explicitly in the introduction, now stating (lines 81-84): "*This pattern suggests that, relative to individuals without apathy, both hypo- and hyper-connectivity in different circuits of the fronto-BG network can be associated with apathy.*" (see response to comment #1 for the full paragraph). We have also adapted the text of the discussion section on this aspect to make it clearer. E.g., lines 637-645: "*At first glance, these findings may seem at odds with part of the literature showing links between both subclinical and clinical apathy and brain abnormalities such as grey matter atrophy^{36,72-74}, hypoactivation⁷⁵, hypometabolism⁷⁶, white matter lesions^{18,72}, and reduced functional connectivity^{27,77,78}. However, as mentioned in the Introduction, prior studies have reported mixed findings, with both hypo- and hyper-connectivity patterns observed in relation to apathy, suggesting that either reduced or excessive connectivity may reflect maladaptive network dynamics contributing to motivational impairments. In fact, based on the literature, such findings of hyperconnectivity may be interpreted in two ways. First,...*".

I may have missed it but do the authors explain the anticipated direction of the short- and long-latency OFC-M1 ppTMS? Does OFC have a facilitatory or inhibitory influence on M1?

Authors' answer: Thank you for this comment. In dual-coil ppTMS studies of long-range OFC-M1 connectivity, the direction of influence (i.e., inhibitory vs. facilitatory) can vary across individuals. This variability is reflected in the present data, where MEP ratios are centered around 1 but span values both below and above 1, indicating mixed influences across participants at rest. We now clarify this point explicitly in the legend of Figure 3B (lines 379-383) and refer to prior work reporting similar inter-individual variability in long-latency OFC-M1 connectivity.

Figure 2 caption p11: "The green bars represent..."

P39: "We performed computational phenotyping..."

P40: "the stronger effort valuation IS..."

Fig S2 caption: "The correlogram (LEFT panel)..."

Authors' answer: Thank you for pointing out these errors. We have corrected them in the revised manuscript.

Reviewer #2 (Remarks to the Author):

In this manuscript, Derosiere and colleagues investigated potential structural and functional changes underlying apathy, focusing on the frontal cortex, primary motor cortex (M1), and basal ganglia. They used MRI-based tractography and transcranial magnetic stimulation (TMS) to examine these regions. The study involved a large group of healthy subjects performing an effort-based decision-making task. The authors correlated structural and effective connectivity data with computational parameters derived from the task and with apathy scores. They found clear correlations between apathy scores and specific circuit connectivity and provided evidence for differential contributions of these circuits to effort versus reward valuation. Overall, I found the manuscript to be very well written and easy to follow. To the best of my knowledge, the results are novel and appear robust due to careful data analysis. I have only minor comments. I would like to disclose that while I am very interested in the mechanisms underlying decision-making and motor control in the context of effort-based tasks, I am not a specialist in human behavior or the imaging techniques used in this study.

Authors' answer: Thank you very much for the overall enthusiasm on our article. We have taken into account each of your suggestions below.

Comments:

#In the introduction, the authors refer to the basal ganglia (BG) in general. However, different regions of the BG have not been equally implicated in effort/reward decision-making. It would be helpful if the authors could discuss this more specifically in the introduction.

Authors' answer: Thank you for this helpful suggestion. We have revised the introduction to specify the distinct contributions of different BG nuclei to effort and reward valuation, including the putamen, nucleus accumbens, globus pallidus, and subthalamic nucleus. The revised text now reads (lines 68-76): "*Whether clinical or subclinical^{17,18}, apathy is tied to alterations, consisting predominantly of decreases¹³, though occasionally of increases, in the volume and activity of distributed fronto-basal ganglia (BG) structures known for their role in deciding to exert efforts for rewards^{13,24,25}. These structures include the supplementary motor area (SMA)¹⁸ and the putamen²⁹, involved in effort valuation³⁰⁻³², the orbitofrontal cortex (OFC)³³, engaged in reward valuation^{34,35}, and the nucleus accumbens^{36,37}, thought to weigh up efforts and rewards^{24,38}. Alterations are likewise observed in downstream BG nuclei, such as the globus pallidus²², where effort and reward computations have also been reported^{39,40}.*"

#For a naive reader, it would be nice to at least provide a couple of sentences regarding intra-M1 GABAa, GABAb, and glutamatergic circuits connectivity can be inferred from such imaging techniques

Authors' answer: We thank the reviewer for this suggestion. We have revised the manuscript to clarify how intra-M1 GABA_A, GABA_B, and glutamatergic circuits are selectively probed using ppTMS, based on well-established literature on the underlying neurophysiological principles. Specifically, we now explain in the introductory paragraphs of the Results section (lines 221-232): “*We also used a single-coil ppTMS approach to examine intra-M1 inhibitory and excitatory circuits. This technique relies on systematic variations of the inter-stimulation interval (ISI) and the intensity of the CS to preferentially probe different circuits with a single coil positioned over M1. Specifically, short ISIs (e.g., 3 ms) with subthreshold CS target fast, low-threshold GABA_A-mediated inhibition, as supported by pharmacological and electrophysiological studies (see⁵⁷, for review). Then, longer ISIs (e.g., 100 ms) with suprathreshold CS engage slow, high-threshold GABA_B-mediated inhibition, while intermediate ISIs (e.g., 10 ms) with subthreshold CS recruit glutamatergic neurons. The functional strength of these specific intra-M1 circuits is quantified by considering changes in the amplitude of MEPs elicited by the test stimulus (TS) when this TS is preceded by a CS, compared to when it is applied alone, providing a measure of intra-M1 effective connectivity^{45,57}.*”

The Methods section (*Single-coil ppTMS* sub-section) of the revised manuscript also include detailed justifications for the chosen ISIs and intensities, with appropriate references. We hope this will make the rationale accessible and clear to both expert and non-expert readers.

#In the results section, there was little practical information about the decision-making task, yet the authors derived effort and reward sensitivities from it and examined their correlation with imaging data. For a naive reader, it would be helpful to provide a clearer description of the task structure in the main text, rather than requiring reading the methods. More generally, I wondered whether it was appropriate to limit the description of this key component to a schematic overview in Figure 1 and Supplementary Figure 2. A more detailed presentation of the behavioral results in the main figures or text might improve comprehension of this study.

Authors' answer: As suggested, we have expanded both the main text and the legend of Figure 1 to include a clearer and more detailed description of the decision-making task, including its structure, trial time course, and how effort and reward levels were manipulated. We also now provide a step-by-step account of the behavioral results, highlighting how effort and reward influenced participants' acceptance rates before diving into the computational modelling of it. We considered adding a separate figure focused solely on behavioral results, but felt this would disrupt the logical flow of the manuscript, which presents the protocols and results sequentially in Figures 1A to 1D. We think that inserting an additional figure at this stage would interfere with this structure and complicate figure referencing. Instead, we opted to integrate the behavioral findings more thoroughly into the main text and Figure 1 legend. Along with the content of Figure 1B, and Supplementary Figures 2A. B. (also added in response to Reviewer 2, see below), and 4, we believe the revised manuscript now offers a much clearer presentation of the task and behavior.

The main text now reads (lines 147-158): “To address this, we employed an effort-based decision-making task, where participants decided whether to exert varying levels of effort with their right arm (biceps contraction) to obtain monetary rewards (see Figure 1.B and Methods, Effort-based decision-making task). Briefly, on each trial, participants were presented with one of 16 possible combinations of effort and reward levels, defined by 4 effort levels (20%, 40%, 60%, or 80% of their maximal voluntary contraction, MVC) and 4 reward amounts (1, 5, 10, or 20 euro cents). Using a keyboard, they had 5 seconds to decide whether to accept or reject the offer. If they accepted, they performed an isometric contraction of the right biceps, monitored via EMG, and received the reward if the contraction met both intensity and duration criteria (see Methods for full details). A key behavioral measure in this task is the acceptance rate, which reflects participants’ willingness to exert effort for reward. As expected, we replicated the classical pattern: acceptance rates decreased with increasing effort and increased with increasing reward (Figure 1B; Supplementary Figure 3).”

And the legend of Figure 1.B (lines 258-273): “**B. Computational phenotyping:** In the effort-based decision-making task (top left), participants decided whether to exert varying effort levels with their right biceps to earn different monetary rewards. Each trial displayed a required effort level (20–80% of MVC) via a vertical gauge, and a monetary amount (1–20 euro cents). Participants had 5 seconds to accept or reject the offer using a keyboard. If they accepted, they attempted an isometric contraction monitored in real-time via EMG. Success required maintaining the target force for at least 2.7 seconds within a 4-second window. Accepted trials were rewarded if performance criteria were met; otherwise, no reward was given. The graph (bottom left) shows mean acceptance rates across effort levels for each reward amount in our sample of 45 subjects (green solid line), illustrating the expected pattern: acceptance increased with reward and decreased with effort (see also Supplementary Figure 3). Black dashed lines indicate the fit of our computational model. Parameters β_{Effort} and β_{Reward} were estimated from the modeling of acceptance rates using the equation presented on top of the graph (see Supplementary Figure 2 for model comparison). Here again, the density distributions on the right depict the high inter-individual variability in β_{Effort} and β_{Reward} values among our group of 45 healthy subjects.”

#Related to the point above, is there no possibility to use movement related data to infer apathy? One could expect that subjects with a higher apathetic score would exhibit “difficulty” in generating the strongest force for the lowest reward. Or does the visual feedback on strength level prevent this kind of effect?

Authors’ answer: We thank the reviewer for this suggestion. Following this comment, we conducted additional analyses to explore whether movement-related data could further inform individual differences in apathy. While the idea of examining behavior in high-effort/low-reward trials is highly relevant, it was unfortunately not feasible in our dataset: the vast majority of participants rejected all of these trials, resulting in too few accepted trials – and thus executed contractions – to support meaningful analysis. Instead, we focused on a subset of conditions with sufficient trial counts across participants (i.e., trials with 20% and 40% MVC and 10 and 20 cent rewards). In these trials, we computed a measure of excess contraction – defined as the amount of contraction voluntarily produced beyond the target level (e.g., producing 35% MVC in

a 20% MVC trial corresponds to 15% excess contraction). This metric provides an index of voluntary drive or spontaneous effort investment during action execution, beyond the minimum required to obtain the reward. We found that excess contraction was significantly negatively correlated with apathy scores (e.g., $R = -0.39$, $p = .007$ in 20% MVC/10 cents trials; see Supplementary Figure 2.B), showing that individuals with higher apathy levels exert less contraction than less apathetic subjects. This result supports the notion that apathy is not only reflected in the willingness to engage in effortful actions, but also in how much effort is invested during execution. We have added this analysis and its interpretation in the main text and Supplementary Figure 2.B.

The revised text reads (lines 165-175): “*In addition, we analyzed a metric of effort exertion, defined as the excess contraction produced beyond the required level (e.g., generating 35% of MVC in a trial with a 20% MVC target corresponds to 15% excess contraction), which provides an index of voluntary drive or overcommitment during effort execution. We found that higher apathy scores were significantly associated with lower excess force (e.g., $R = -0.39$, $p = .007$ in 20% MVC/10 cents trials; see Supplementary Figure 2.B), indicating reduced spontaneous effort investment. Together with the acceptance behavior results, these findings support the value of effort-based decision-making tasks for capturing inter-individual differences in motivational drive across both the decision and execution phases of goal-directed behavior.*”

Here is Supplementary Figure 2:

Supplementary Figure 2: Apathy scores correlated negatively with decision-making task metrics reflecting motivational drive during both decision and execution phases.

A. Individuals with higher apathy scores presented a lower acceptance rate. The correlogram (right panel) displays Pearson correlation coefficients between apathy scores and acceptance rates across different combinations of effort and reward levels. Blue shades indicate negative correlations; stars indicate statistical significance ($p < .05$). A significant effect was observed for instance in trials with high effort (80% MVC) and high reward (20 cents), where a negative correlation was found ($R = -0.29$, $p = .046$; middle panel). This indicates that participants with higher apathy scores were less likely to accept these high-reward offers, suggesting reduced reward sensitivity. This relationship must be interpreted with caution due to ceiling effects, as several participants accepted nearly all high-reward trials (acceptance rate ~ 1). Nevertheless, the top left of the scatterplot illustrates that higher apathy scores were associated with lower acceptance in these trials. To further illustrate this effect, participants were split into low- and high- apathy groups (first and last quartiles, $n = 10$ per group). Low- apathy participants accepted 92% of these high-reward/high-effort offers (0.92 ± 0.04), while high- apathy participants accepted only 68% (0.68 ± 0.11). The acceptance rate of the high- apathy group was significantly lower than 1 ($t_9 = -2.78$, $p = .021$), and the between- group difference was marginally significant ($t_{18} = 1.96$, $p = .075$). These results suggest that higher apathy scores may be associated with a blunted behavioral drive to obtain high rewards in healthy subjects.

B. Individuals with higher apathy scores presented a lower excess contraction during effort production. We analyzed a metric of excess contraction, defined as the amount of biceps contraction voluntarily produced beyond the required target (e.g., producing 35% MVC in a 20% MVC trial corresponds to 15% excess contraction), as an index of spontaneous effort investment. This analysis focused on a subset of trials with sufficient accepted responses across participants (20% and 40% MVC targets combined with 10 and 20 cent rewards). The correlogram (left panel) displays Pearson correlation coefficients between apathy scores and excess contraction across these conditions. Blue shading indicates negative correlations; asterisks denote statistical significance ($*p < .05$, $**p < .01$). Significant negative correlations were observed in all four effort-reward conditions (middle panel shows correlation for 20% MVC / 10 cents: $R = -0.39$, $p = .007$), indicating that participants with higher apathy levels produced smaller excess contractions. Single-trial examples from two representative participants with low vs. high apathy scores (S33 and S11, respectively) are also shown (left panel). Together with the acceptance rate findings, these results demonstrate that apathy is reflected not only in reduced willingness to engage in effortful actions but also in reduced voluntary effort investment during movement execution.

#One intriguing result is that the computational variables derived from the decision making task do not correlate with apathy scores. Could it be that the task used in the lab does not require sufficient effort or involve sufficiently vital rewards? This could be discussed further.

Authors' answer: We thank the reviewer for this comment, which we now explicitly address in the Results section of the revised manuscript.

The text now reads (lines 185-198): *“Nonetheless, neither β_{Effort} nor β_{Reward} significantly correlated with apathy scores in our sample. Bayesian regression analyses provided moderate evidence in favor of the null hypothesis ($BF_{01} = 2.97$ for β_{Effort} ; $BF_{01} = 2.24$ for β_{Reward} ; see Supplementary Figure 3.D). One possible explanation is that the effort and reward manipulations in this controlled laboratory task, although widely validated and commonly used to probe motivational processes, may not fully capture the complexity or ecological relevance of the cost-benefit trade-offs assessed with apathy questionnaires. As such, the latter relate more directly to complex, real-world goal-*

directed behaviors than to more simple decisions probed in laboratory tasks. Another explanation is that the model fitting process, by design, emphasizes latent constructs across trial types, averaging out trial-to-trial variability in acceptance rates that might reflect subtle motivational differences linked to apathy, particularly in conditions like high-reward trials (as shown in Supplementary Figure 2.A). This may limit the detection of significant correlations between apathy scores and model parameters. Still, model-derived parameters offer a robust framework to describe general effort and reward valuation processes, and are thus widely used in the field to capture inter-individual differences in these processes^{2,4-6,10,11,56}.”

Typos:

In the method section: As such, in the above equation, the more negative the β_{Effort} is, the stronger effort valuation *iss*

Supplementary Figure 1: the last sentence of the legend is incomplete.

Authors' answer: Thank you for pointing out these errors. We have corrected them in the revised manuscript.

Reviewer #3 (Remarks to the Author):

This paper explores the neural correlates of apathy and effort-based decision-making in healthy individuals, focusing on connectivity within fronto-basal ganglia (BG)-motor circuits. Using a multimodal approach combining diffusion-weighted MRI tractography and paired-pulse transcranial magnetic stimulation, the researchers assessed both structural and effective connectivity among 45 participants. Behavioral data were collected via an effort-based decision-making task, analyzed with computational modeling to extract sensitivity to effort (β_{Effort}) and reward (β_{Reward}). Apathy levels were assessed with validated questionnaires. They found that apathy scores positively correlate with both structural connectivity (in SMA-M1, SMA-putamen, and OFC-caudate tracts) and effective connectivity (in long-latency OFC-M1 circuits). Effort sensitivity was linked to structural connectivity in SMA-M1 and SMA-NAcc pathways but not to effective connectivity. Conversely, reward sensitivity was associated with structural connectivity in the NAcc-GP tract and effective connectivity in intra-M1 GABAergic circuits. Interestingly, the study found no correlation between apathy or valuation measures and M1 net output, suggesting that connectivity—rather than output strength—is the key driver. The manuscript is well written and methodologically thorough, with findings highlighting the distinct fronto-motor circuits underlying apathy, effort, and reward valuation, and potential implications for targeted neuromodulation strategies in clinical populations. However, before considering this work for publication, there are some theoretical and methodological issues that should be addressed.

Authors' answer: Thank you very much for the overall positive opinion on our article and the study. We have addressed each of the potential theoretical and methodological concerns below.

My main concerns include interpretations of the results of the one side and the model fitting on the other, plus some minimum polishing/minor points. 1. Authors present an interesting study of the apathy phenomenon as a network-level-phenomenon involving fronto-basal ganglia circuits and their connectivity to motor regions. However, several clarifications and enhancements could strengthen the framing of the study, still partially lacking clarity and motivation. First, the construct of apathy is introduced as a unitary concept, whereas the literature consistently distinguishes between behavioral, emotional, and cognitive subtypes (for instance Levy & Dubois, 2006 <https://doi.org/10.1093/cercor/bhj043>; Radakovic & Abrahams, 2014; <https://doi.org/10.1016/j.psychres.2014.06.010>). Given the use of an effort-based decision-making task, it would be helpful for the authors to explicitly situate their work within the framework of behavioral or motivational apathy and reference prior work that supports this dimensional view.

Authors' answer: We thank the reviewer for this comment. In response, the revised introduction now explicitly situates our work within the dimensional model of apathy, hence distinguishing between behavioral, emotional, and cognitive components, following the work of Levy & Dubois (2006) and Radakovic & Abrahams (2014). In line with this, we have conducted an additional set of analyses using apathy subscores to

determine the specificity of our connectivity findings with respect to these distinct apathy dimensions. These new analyses revealed that both structural and effective connectivity measures involving the OFC – particularly the OFC-dCaudate tract and the long-latency OFC-M1 circuit – were significantly and selectively associated with behavioral apathy, but not with emotional or cognitive components. In contrast, connectivity in the SMA-dIPutamen tract covaried more specifically with emotional apathy. These results support the hypothesis that fronto-striato-motor pathways may differentially contribute to specific facets of apathy, and in particular, that long-latency circuits linking OFC to motor regions are preferentially involved in the behavioral dimension. We have added these results as a new supplementary figure (Supplementary Figure 5) and detailed the corresponding results in the revised manuscript.

The text now reads:

1. In the Introduction (line 66): *“It has become clear that apathy is a multidimensional syndrome^{22,23} and is not attributable to a single brain region but can rather be seen as a network disorder^{13,18,24–28}.”*
2. In the Results section (lines 291-307): *“The regression identified three tracts, namely SMA-M1, SMA-dIPutamen, and OFC-dCaudate, with non-zero coefficients, suggesting potential correlation with apathy scores (LASSO’s β coefficients = 0.001, 0.00005, and 0.016, respectively; Figure 2.A). Partial correlation analysis confirmed significant positive correlations between apathy scores and the number of streamlines in these circuits, even when controlling for the potentially confounding variables ($R = 0.34$, $p = .028$; $R = 0.33$, $p = .033$; and $R = 0.39$, $p = .009$, respectively; Figure 2.B), indicating that higher apathy scores are associated with stronger structural connectivity in these circuits. Interestingly, a second-level analysis of apathy subscores revealed that streamline count in the OFC-dCaudate tract was preferentially associated with the behavioral dimension of apathy (i.e., assessed through the action initiation subscore of the apathy; $R = 0.31$, $p = .015$; Supplementary Figure 5.A), but not with emotional or cognitive dimensions of apathy ($R = -.002$, $p = .998$, and $R = 0.13$, $p = .405$, respectively), whereas connectivity in the SMA-dIPutamen tract covaried more specifically with the emotional component ($R = 0.32$, $p = .045$). These findings suggest that the observed associations with global apathy scores may reflect distinct contributions of fronto-striatal circuits to specific apathy dimensions.” AND “This included ratios for short- and long-latency SMA- and OFC-M1 circuits, as well as intra-M1 GABAa, GABAb, and glutamatergic circuits. The regression revealed two circuits (short- and long-latency OFC-M1) with non-zero coefficients (LASSO’s β coefficients = -0.23 and 0.61, respectively; Figure 3.A). Importantly, partial correlation analyses confirmed a significant positive correlation for the long-latency OFC-M1 circuit ($R = 0.38$, $p = .014$; Figure 3.B), indicating that higher apathy scores are associated with a stronger facilitatory influence of OFC on M1 (i.e., MEP ratios > 1) in this long-latency circuit. However, this was not the case for the short-latency OFC-M1 circuit ($R = -0.09$, $p = .571$; Supplementary Figure 5). Interestingly, the significant positive correlation for the long-latency OFC-M1 circuit was specific to the behavioral dimension of apathy ($R = 0.37$, $p = .015$), but not to the emotional or cognitive components ($R = 0.22$, $p = .152$, and $R = 0.19$, $p = .222$, respectively; Supplementary Figure 5.B). This mirrors the similar association found for OFC-dCaudate structural connectivity, suggesting that disruptions in OFC-NG-M1 circuits may selectively contribute to reduced action initiation.”*

Here is supplementary Figure 5:

Supplementary Figure 5: Apathy subscores covary preferentially with connectivity in specific fronto-striatal and fronto-motor circuits. A. Structural connectivity. The leftmost columns reproduce the main results presented in the manuscript, showing partial correlations between global apathy scores and streamline counts in two tracts identified by LASSO regression (OFC-dCaudate and SMA-dIPutamen). The three right columns present the same partial correlation analyses separately for behavioral, emotional, and cognitive apathy subscores, derived from the LARS-e questionnaire: Action Initiation subscale (behavioral apathy), Emotional Response subscale (emotional apathy), and Intellectual Curiosity – Novelty subscale (as a proxy for cognitive apathy). These analyses revealed that streamline count in the OFC-dCaudate tract was specifically associated with behavioral apathy ($R = 0.31$, $p = .049$), but not with emotional or cognitive apathy ($R = -0.002$, $p = .988$; $R = 0.13$, $p = .405$, respectively). In contrast, connectivity in the SMA-dIPutamen tract was positively associated with emotional apathy ($R = 0.32$, $p = .045$). SMA-M1 connectivity showed no significant association with any apathy subscore, suggesting that its contribution to global apathy may not reflect any single dimension in isolation. These findings support the notion that distinct fronto-striatal pathways are differentially involved in specific components of apathy. **B. Effective connectivity.** As observed for structural connectivity in A, partial correlation analysis for the long-latency OFC-M1 circuit showed a significant association with behavioral apathy ($R = 0.37$, $p = .015$), but not with emotional ($R = 0.22$, $p = .152$) or cognitive apathy (R

= 0.19, $p = .222$), further implicating long-latency fronto-striato-motor pathways in deficits in action initiation. Note: Variables included in these analyses were selected via LASSO regression, reducing the likelihood of false positives; however, p-values reported here are uncorrected for multiple comparisons and should be interpreted with caution.

Second, while the authors argue that apathy varies in healthy individuals, they could more clearly articulate the relevance of studying subclinical apathy by citing empirical work on its neural correlates in non-clinical populations (like Bonnelle et al., 2015 <https://doi.org/10.1093/brain/awv278>; Grogan et al., 2018, <https://doi.org/10.1016/j.neuropsychologia.2018.05.008>). Additionally, the rationale for including M1 in the apathy network is compelling but underdeveloped; the introduction would benefit from theoretical grounding—such as distributed decision-making or embodied cognition models—to explain how M1 might contribute to motivational processes beyond simple movement execution (Cisek & Kalaska, 2010, <https://doi.org/10.1146/annurev.neuro.051508.135409>). Finally, the authors may consider refining the final paragraph of the introduction to better emphasize the specific research questions, reduce detailed methodological content, and more clearly highlight the novel contribution of their multimodal approach. Together, these revisions would sharpen the conceptual framing and improve clarity for a broader audience.

Authors' answer: We have now incorporated most of the suggested references on subclinical apathy (Bonnelle et al., 2016, Cerebral Cortex) and on decision-making considered in an embodied cognition framework supporting the theoretical rationale for including M1 (Cisek & Kalaska, 2010). Finally, we have revised the final paragraph of the Introduction to better articulate our research questions, reduce methodological details, and more clearly highlight the added-value of our multimodal approach and the main findings (lines 107-133): *“In this study, we aimed to characterize how structural and effective connectivity within fronto-striato-motor circuits relate to individual differences in apathy in the general population. Specifically, we asked whether distinct circuits are differentially associated with apathy levels, as measured through questionnaires^{18,19}, and the valuation of efforts and rewards, as measured through computational phenotyping of behavior in an effort-based decision-making task^{10,11,60,61}. While previous studies have linked apathy to altered connectivity within fronto-BG networks, findings have been mixed in terms of directionality – reporting both increases and decreases in connectivity – and largely focused on non-motor circuits. In light of these mixed findings and the paucity of data on motor circuits¹⁸, the present study hypothesized altered connectivity within these circuits, without making directional assumptions regarding connectivity–behavior relationships. To address these questions, we combined DWI and dual-coil ppTMS to probe both structural and effective connectivity in key fronto-M1 and fronto-BG-M1 circuits originating from the SMA and OFC. In addition, single-site ppTMS was used to assess the physiological properties of intra-M1 circuits influencing corticospinal output. This multimodal framework allowed us to capture connectivity patterns across both anatomical and effective levels, from frontal input to motor cortex, and extending to the final corticospinal output stage. Among the most robust findings, we observed that both OFC-BG structural connectivity and OFC-M1 long-latency effective connectivity were positively associated with apathy scores, thus replicating this relationship across our two connectivity measures. Moreover, SMA-M1 structural connectivity was related to*

both apathy and effort valuation, consistent with SMA's known role in effort processing^{30,31}. Altogether, these findings suggest that distinct fronto-motor circuits – originating from both OFC and SMA – contribute to apathy and effort valuation, highlighting complementary circuits as potential neuromodulatory targets for specific mechanisms of disrupted motivated behavior.”.

2. While the study compellingly demonstrates behavioral correlations between apathy scores and task performance in high-effort/high-reward trials, the lack of significant associations between apathy and the model-derived β Effort and β Reward parameters raises important concerns about the sensitivity or specificity of the current computational model. This disconnect is particularly noteworthy given the central goal of the study—to characterize apathy in terms of latent motivational constructs. The reliance on static, two-parameter models (e.g., β Effort and β Reward) may oversimplify the decision-making process, especially when applied to a heterogeneous construct like apathy that likely involves fluctuating cognitive and affective states (e.g., fatigue, learning, internal variability in valuation). As such, while the current modeling approach follows precedent, it might not capture the full complexity of motivational dynamics in real-time decision-making. Authors should consider incorporating additional parameters into the computational model that reflect decision noise or exploration/exploitation tradeoffs (e.g., softmax temperature or lapse rate); effort discounting curves (nonlinear or parabolic rather than linear); fatigue accumulation or trial-wise adaptation (dynamic models with trial history). Also, they could provide a post-hoc mediation by examine whether task behavior mediates the relationship between apathy scores and connectivity, which might clarify whether the lack of correlation with β parameters is a modeling artifact or a real dissociation.

Authors' answer: We thank the reviewer for raising these interesting insights. We indeed compared six models that differed in their cost functions (linear, quadratic, or exponential) and in the presence or absence of a β_{Time} parameter capturing trial-wise adaptation (*i.e.*, fatigue accumulation or learning effects). The winning model, used throughout the manuscript, included a quadratic cost function and a β_{Time} parameter, thus capturing both non-linear effort discounting and temporal dynamics in valuation, as suggested by the reviewer. Details of model selection, fit quality, and subject-level comparison are included in Supplementary Figure 3 of the revised manuscript and the Methods section (lines 1132-1154): *“Based on previous literature^{8,10,117}, we tested candidate models of value computation, using standard methods (minimization of Bayesian information criterion (BIC) and visual inspection of model fits, see Supplementary Figure 3). More specifically, for each trial, the model computes a subjective value of the offer that depends on the proposed effort and reward. This value is then transformed to an estimated probability of acceptance (between 0 and 1) based on a softmax function⁸. Parameter estimates were determined with the function `fmincon` on Matlab 2022b (Matlab, Mathworks, USA) with the goal to minimize the difference between each participant's modelled and real choice data (*i.e.*, to minimize the negative log-likelihood of the model given the data). **The models differed in the form of the cost function (linear, quadratic, or exponential effort discounting) and in the inclusion or absence of a β_{Time} parameter capturing trial-wise adaptation (*i.e.*, modulation of value computation by trial number). The winning model was the***

following: $Value = \beta_{Effort} \times EFFORT_{LEVEL}^2 \times (1 + \beta_{Time} \times t) + \beta_{Reward} \times REWARD_{LEVEL} + \beta_0$
Here, the value of an offer depends on a quadratic function of the proposed $EFFORT_{LEVEL}$ and on the available $REWARD_{LEVEL}$ (modulated by the β_{Effort} and β_{Reward} parameters, respectively). To account for potential time-on-task effects on value computation (due to the repetition of trials), this model also includes a β_{Time} parameter that linearly modulated the cost term based on the trial number¹⁰.”

Regarding the proposed mediation analysis, as suggested, we investigated whether task behavior mediated the relationship between apathy scores and connectivity. While apathy scores were significantly correlated with acceptance rates in specific trial types (Supplementary Figure 2.A), these behavioral measures did not correlate with structural or effective connectivity in the circuits identified (e.g., acceptance vs. OFC-dCaudate $p = .170$; SMA-dIPutamen $p = .182$; SMA-M1 $p = .978$; effective connectivity in OFC-M1 $r = .937$). This precludes a formal mediation analysis. Given the absence of behavioral-connectivity correlations, we chose not to include these null results in the main text to avoid overcomplicating an already extensive set of analyses. We believe our revised manuscript now more accurately conveys the scope and limitations of the computational modeling approach.

3. When authors consider the past two decades of “progress in neuroimaging and brain stimulation have enabled new ways to explore this idea in humans. Advanced analyses of diffusion weighted imaging (DWI) such as streamline tractography allow quantifying how structural connectivity in specific circuits varies among individuals^{48,49}. Additionally, paired-pulse transcranial magnetic stimulation (ppTMS) allows pre-activating certain circuits and measuring the impact of this pre-activation on neural activity in M1's corticospinal neurons^{38,50,51}, providing a measure of how effective connectivity in circuits projecting to these neurons varies among individuals. Integrating these complementary methods through a multimodal framework thus provides a unique opportunity to investigate the different circuits connecting SMA, OFC and the BG to M1 in a comprehensive way” they missed some recent literature that already started exploring the DWI-TMS combination in the context of action decision and reviewing network including SMA and other frontal and premotor areas (Tagliaferri et al., 2023 <https://doi.org/10.1523/JNEUROSCI.0406-23.2023>, Tagliaferri et al., 2024 <https://doi.org/10.1007/s00429-024-02778-4>)

Authors' answer: We thank the reviewer for this relevant suggestion. We now refer to Tagliaferri et al. (2023; unless we are mistaken, the paper in 2024 does not refer to work combining DWI and TMS) and acknowledge their contribution to the emerging literature combining DWI and TMS to study network-level mechanisms of action control (lines 102-104): “*As such, recent studies have begun combining DWI and TMS to investigate cognitive processes⁵⁸. Building on this emerging multimodal framework, integrating these complementary approaches provides a unique opportunity to comprehensively investigate the circuits connecting SMA, OFC, and the BG to M1.*”

Finally, author measure apathy via self-report (LARS-e), which may not fully capture

behavioral or neurobiological manifestations. A growing consensus suggests combining self-report tools like LARS-e with behavioral paradigms (e.g., EEfRT, grip-force tasks - 10.1001/jamapsychiatry.2017.2901; 10.1038/s41583-018-0029-0) and EMA (10.1037/abn0000191; 10.1016/j.schres.2021.02.005) to provide a more accurate, multidimensional measure of apathy. This triangulation can help disentangle motivational deficits from reporting biases or mood symptoms. I suggest including some objective behavioral tasks or ecological momentary assessments to complement the questionnaire, and as a minimum to incorporate the limitations of this approach.

Authors' answer: We thank the reviewer for this comment and now mention this in the Methods (lines 1075-1090): *"We utilized the extended version of the Lille Apathy Rating Scale (LARS-e), adapted for a healthy population by Bonnelle et al. (2015). The LARS-e consists of 51 items, with 5 propositions for each item, and grades ranging from 1 to 5 for each item, reflecting the subject's level of motivation (a higher number indicating higher motivation). The apathy score is calculated by subtracting the score of each item from the maximum score of 5, then summing up the corresponding items. The obtained sum is divided by the number of items taken, resulting in a mean score between 1 and 5, with a higher score indicating a higher level of apathy¹⁸ (see Data processing section below). While recent work has proposed the construction of composite trait apathy scores using factor analysis across multiple questionnaires¹¹⁶, a large and robust body of literature currently uses single-scale scores alone as a validated and sensitive measure of apathy-related motivational states, particularly when combined with task-based behavioral assessments to isolate specific motivational components. Our study followed this well-established approach to directly relate inter-individual variability in apathy to structural and effective connectivity, while acknowledging that no single method can capture the full complexity of motivational processes."*

Minor suggestion/improvement: as I suggested in point 1, while M1 involvement is novel and well-justified, in the end the authors might have overemphasized its role without considering broader network contributions in the explanation of the results beyond the tractography. They could take into deeper consideration functional connectivity or broader fMRI networks (e.g., default mode or salience networks) that may also contribute to apathy.

Authors' answer We now discuss this point before concluding the Discussion section, as follows (lines 742-749): *"While our study focused on specific fronto-BG circuits projecting to M1, apathy is widely recognized as a network-level disorder. Beyond fronto-BG circuits, structural and functional MRI studies have shown alterations in large-scale networks, including the default mode and salience networks¹⁰²⁻¹⁰⁴. Our targeted approach was designed to test mechanistic hypotheses about the contribution of motor circuits to apathy, as well as to effort and reward valuation. Nonetheless, future studies incorporating whole-brain structural and/or functional connectivity analyses could complement our work and help clarify how these motor pathways interact with broader large-scale networks."*